# Unpiloted Aerial System (UAS)-Supported Biogeomorphic Analysis of Restored Sierra Nevada Montane Meadows

**Jerry Davis [1,](ORCID), Leonhard Blesius [1], Michelle Slocombe [2], Suzanne Maher [3], Michael Vasey [4], Peter Christian [5] and Philip Lynch [1]**

[1] Department of Geography & Environment, San Francisco State University, San Francisco, CA 94132, USA; lblesius@sfsu.edu (L.B.); plynch@mail.sfsu.edu (P.L.)
[2] CalRecycle, Sacramento, CA 95814, USA; michelle.slocombe@calrecycle.ca.gov
[3] Chabot College, Hayward, CA 94545, USA; smaher@Chabotcollege.edu
[4] SF Bay National Estuarine Research Reserve, San Francisco, CA 94920, USA; mvasey@sfsu.edu
[5] HERE Technologies, Berkeley, CA 94704, USA; peter.christian@here.com
\* Correspondence: jerry@sfsu.edu

**Abstract:** The benefits of meadow restoration can be assessed by understanding the connections among geomorphology, hydrology, and vegetation; and multispectral imagery captured from unpiloted aerial systems (UASs) can provide the best method in terms of cost, resolution, and support for vegetation indices. Our field studies were conducted on northern Sierra montane meadows (with ≤70 km$^2$ watershed area). The meadows exist in various stages of ecological restoration. Field survey methods included GPS + laser-leveling channel survey, cross-sections, LiDAR, vegetation sampling, soil measurements, and UAS imaging. A sensor captured calibrated blue (465–485 nm), green (550–570 nm), red (663–673 nm), near infrared (NIR) (820–860 nm), and red-edge (712–722 nm) bands at 5.5 cm resolution (as well as thermal at 81 cm resolution) and provided multispectral images and derivative vegetation indices such as the normalized difference vegetation index (NDVI) and red-edge chlorophyll index (Cl$_{re}$). This fine-scale imagery extended our morphometric assessment of post-restoration channel bedform patterns and sinuosity related to *Carex*-influenced soil properties and *Salix* influence, and also documented groundwater-related effects via *Carex* patterns evident from spring snowmelt images, as well as NDVI and Cl$_{re}$ (derived from spring and summer images) in growing to senescent phenological stages. *Carex* was significantly associated with low bulk density and high soil moisture, NDVI, and Cl$_{re}$ in low-lying areas, and channel sinuosity was significantly associated with willow influence. Our methods can be applied by restoration managers to assess where projects are threatened by renewed incision and to document levels of carbon sequestration significant to addressing climate change.

**Keywords:** UAS; UAV; meadow; NDVI; geomorphology; restoration; vegetation; hydrophilic; change detection; LWIR; thermal imaging

## 1. Introduction

The study of alpine and montane meadows lies at the nexus of an array of environmental sciences including hydrology, ecology, edaphology, geomorphology, and micrometeorology; and understanding the interactions among these systems is key to understanding their response to restoration efforts. Wet meadows, in particular, have attracted increasing attention due to their significance related to threatened species, water resources, and the carbon cycle [1,2], and efforts to restore meadow ecosystem services have expanded in recent decades [3,4]. Discerning effects of restoration efforts amidst the

biologically and geomorphologically dynamic systems of meadow channels can be achieved through the fine resolution afforded by unpiloted aerial systems (UASs) employing multispectral cameras designed for precision agriculture.

The signal of meadow restoration can come in many forms and at multiple spatial and temporal scales of analysis, and thus imagery from various sources ranging from satellite platforms to handheld photography plays an important role depending on the requisite environmental variables and their spatial and temporal scales. Silverman et al. [5] demonstrated the value of freely available Landsat 30 m imagery for providing frequent assessment of vegetation indices in restoration projects. Sentinel-2 imagery at 10 and 20 m resolution has also shown considerable promise for interpreting grasslands [6]. Although not freely available, low-cost imagery from new constellations of small satellites such as CubeSats from Planet [7] provides four-band 3 m data on mostly a daily basis. These imagery sources provide high temporal resolution and ample spectral data for assessing the signal of meadow restoration coming from changes in vegetation health and hydrologically defined communities, with a transition to more hydrophilic species being a common goal of restoration, and these changes can often be seen even at these relatively coarse resolutions.

In this study, we focus on the finer scale of detection for the measurement of finer channel characteristics such as width, length, sinuosity, and related vegetation effects where drone platforms provide an advantage in resolution, especially for assessing smaller ephemeral channels which require careful monitoring after restoration. Our study focuses on montane meadows with limited watersheds drained predominantly by ephemeral streams or meadow aquifers, except where degraded by incised channels. We employ topographic measurements and surface sampling coupled with UAS methods for channel measurement and assessment of variables affecting channel development, as part of an evaluation of ecohydrological response to various forms of environmental restoration. Meadow ecohydrologic degradation is tied to geomorphic degradation (vertical channel incision due to greater erosion than deposition) which is driven by stream power, thus, understanding the development of even small ephemeral channels, often with less than a meter width, is important for assessing the ongoing success of restoration projects.

UAS methods for studying vegetative cover have experienced dramatic growth in recent years, with technology driven by the benefits seen in precision agriculture, where fine-scale multispectral images and vegetative indices provide high-frequency views of moisture and nutrient availability stress [8–10]; and recreational UAS use has greatly expanded the availability, affordability, and ease-of-use of remote-piloted and automated platforms. Increasingly available multispectral cameras employing narrow (10–40 nm width) panel-calibrated spectral bands of blue, green, red, near infrared (NIR), and the "red-edge" (RE) near 700 nm provide the ability to create an array of indices useful for interpreting vegetative health. For example, the normalized difference vegetation index (NDVI) and the red-edge chlorophyll index ($Cl_{re}$) have been shown to effectively estimate chlorophyll content and $CO_2$ flux in crops [11,12], and thus help to inform assessment of vegetation health and the carbon cycle in meadows. They can be calculated as follows:

$$NDVI = \frac{(NIR - Red)}{(NIR + Red)} \qquad (1)$$

$$Cl_{re} = \frac{NIR}{RE} - 1 \qquad (2)$$

The deployment of multispectral cameras on UAS platforms has led to an expansion of use in ecological [13] and other environmental science research [14] at a low cost as compared with field surveys or piloted aircraft. The number of applications continues to increase and provides alternatives to more costly approaches (e.g., LiDAR or direct measurement) for estimating vegetation growth rates and biomass [15] or for estimating evapotranspiration [16].

The meadows we see today in the Sierra Nevada have been impacted by a history of livestock grazing and timber extraction, as well as rail and road construction. Sheep and cattle grazing in

the Sierra Nevada was widespread as of the Gold Rush, peaked in 1876, and resulted in meadow degradation with documented gullying [17,18]. In the 19th and early 20th centuries, timber extraction was limited to areas in close proximity to settlements or rail access. The area that later became the Tahoe National Forest experienced a significant impact of feeder railway lines built through montane meadows owing to the ability to connect to the Southern Pacific Railroad [18]. Railroad construction was also able to take advantage of easy grades along riparian areas in the meadows [19].

The meadows display complex feedback between natural systems and anthropogenic intervention [18]. Due to their significant history as grazing lands in the American West, montane meadows have experienced intentional, as well as accidental, alterations of groundwater levels and related channel developments. Variations of groundwater, which influence vegetation alliances [20] from hydrophilic to xeric, interact with soils and biogeomorphic systems, and these systems are significant elements of the biogeochemical carbon cycle [21]. A shift to conditions that are unfavorable to certain plant species leads to their decline, and the biodiversity loss negatively affects local wildlife dependent on those species for habitat. While meadows are highly diverse in character [22], the vegetation, hydrology, geomorphology, and soils of these systems are clearly interlinked [17,23].

A prominent signal of a degraded meadow due to erosion is the lowering of the water table and conversion of plant communities from hydrophilic to more xeric types [17,24], as can be seen in the replacement of sedge (*Carex*) species that mature under seasonal water cover [22] with more mesic grasses and forbs, in many areas extending to xeric plants such as sagebrush (*Artemisia tridentata*). This shift prevents hydrophilic sedge (*Carex*) and rush (*Juncus*) species from creating dense root mats which provide resistance to bank erosion [25–27]. Once initiated, headcuts extend these effects into upstream meadows as gullies. Discontinuous and continuous gullies have been described in relatively steep sites influenced by piping [28] to gradients as low as 1% [29,30]. While many of these systems are in thinly vegetated floors of arroyos [29], processes such as sapping which leads to eroding root mats have also been described in the Sierran meadows [17].

Meadow protection and restoration efforts have been driven by a combination of factors and diverse stakeholders, and in the northern Sierra Nevada have involved the Plumas and Tahoe National Forests involving U.S. Forest Service hydrologists in cooperation with non-governmental organizations such as the Plumas Corporation [31], American Rivers [32], The Sierra Fund [33], and local ranchers on private lands or with grazing leases. Meadow streams provide critical habitats for Lahontan cutthroat trout (*Oncorhynchus clarkii* henshawi), Eagle Lake rainbow trout (*O. mykiss* aquilarum), and California golden trout (*O. mykiss* aguabonita), as well as nonnative trout species. They also provide critical habitats for wetland indicator species such as Sierra yellow-legged frog (*Rana sierra*) [34] and willow flycatcher (*Empidonax traillii*) [35]. Herbaceous cover (as opposed to sagebrush) is clearly favored by sheep and cattle ranchers, as well as wildlife. Healthy meadows can be a carbon sink. With the snowpack in the Sierra projected to decline in the coming years due to climate change [21], meadows are also seen as important for flood storage [4] and water quality [36] because healthy meadows maintain large quantities of winter runoff underground and maintain release throughout the summer.

Approaches to protecting meadows from overgrazing have included exclosure fencing to keep cattle away from riparian areas [37], with grade control structures such as check dams used to arrest gully headcuts [38]. These structures, however, often fail, with erosion laterally bypassing check dams, leading to consideration of other methods such as "pond-and-plug" installations [39,40]. This design forces surface flow away from formerly incised paths to other parts of the meadow, sometimes old unincised channels, and are intended to raise the water table while minimizing stream power in order to avoid renewed incision. There has also been a recent surge of interest in "employing" beavers or beaver dam analogues in meadow restoration [41,42], since a strong argument can be made that beaver dams were once a major control on limiting channel incision and storing sediment [41,43], and even the inherent messiness of beaver dams and regular failures have been simulated by induced meandering methods [44]. Since both xeric and hydrophilic vegetation respond to groundwater changes, a successfully raised water table can be assessed by looking at changes in vegetation species

and seasonal phenology. However, other measures can be used to improve or restore the health of wet meadows, including adaptive management policies specifically targeted at the dominant grazing livestock. The specific type of restoration or improvement strategy needs to be evaluated individually based on meadow health, size, or ecological function [3].

A common goal of meadow restoration is to raise the water table, but this goal is affected by local conditions related to the interaction between aquifers developed in meadow alluvial sediments and geologic aquifers. Seasonal changes in the distribution and amount of soil moisture and evapotranspiration loss through meadows largely depend on these conditions in play for a given meadow, and in turn, influence vegetation patterns and surface flows. While the nature of incised channels and their influence on riparian vegetation is generally well understood, post-restoration reoccupied ephemeral channels can have markedly different relationships with vegetation, and their condition should be the focus of ongoing monitoring. Ephemeral channels in wet meadows can differ significantly in their biogeomorphic properties from alluvial streams [45–47] with, for instance, a greater variance in curvature ratio to channel width and a distinct pattern of bedforms resulting from plunge pool development [48]. The tendency toward developing discontinuous gullies and their potential for growth into continuous gullies points to the need to carefully assess their change over time. The association of channel bedforms with hydrophilic vegetation, such as sedges, furthermore, points to the need to assess vegetation changes. Occupation of old channels and changes to hydrophilic vegetation species are an indication of the success of restoration, and changes in either are a signal of evolutionary changes in the restored system.

Our initial channel research in the northern Sierra meadow systems [48] pointed to the need to understand a greater sample of channels, as well as their vegetation and soil associations, by looking at larger areas of such meadows than could be practically achieved by field survey methods alone. Sedge species such as *Carex utriculata* and *C. nebrascensis* have been seen to typify the cover for undisturbed ephemeral channels. Associations of these sedges with ephemeral flow areas at the lowest elevations in the cross-section of a meadow suggest that channel development could be assessed by detecting vegetation and scouring signals using low-altitude multispectral imagery. In this research, we combine UASs and field methods to examine the signal of hydrophilic plants, such as sedges (*Carex*) and riparian trees and shrubs such as willows (*Salix*), on meadow groundwater levels and their relationship to channel bedforms in longitudinal and lateral dimensions.

## 2. Study Sites, Materials, and Methods

The Carman Creek system of meadows, including Knuthson Meadow (Figure 1), near the northern boundary of Tahoe National Forest in the Feather River watershed, has been the subject of multiple phases of pond-and-plug restoration [49]. Upstream of Knuthson Meadow, especially on West Carman Creek, are several other projects ranging from check dams to pond-and-plug installations. Carman Valley on this west fork exhibits several discontinuous gullies deemed insufficiently degraded to warrant attention more effective than check dams, the more traditional method focused on addressing head cuts. Knuthson Meadow provides a clear example of a shift in surface drainage away from the formerly incised creek, now occupied by a series of ponds and plugs, into numerous shallow channels along the more southerly parts of the meadow, with most of the flow incorporated into the meadow aquifer. Note that these Knuthson channels are substantially smaller than the pre-restoration channel which had grown to several meters in width. The upstream unrestored Carman Valley channels are currently wider despite having a much smaller watershed (see Table 1); here, the more traditional check dams are proving to be ineffective at addressing headcuts, and therefore channels are deeper and wider. Other meadow systems that are the focus of this study include the following: (1) another pond-and-plug restoration project, Dry Creek meadow in the Truckee River drainage; (2) Loney Meadow in the Yuba River watershed, which is being restored using a limited (no excavation) pond-and-plug method [50]; and (3) the perennial Dixie Creek, part of the Red Clover Valley system, being restored using beaver dam analogues [33].

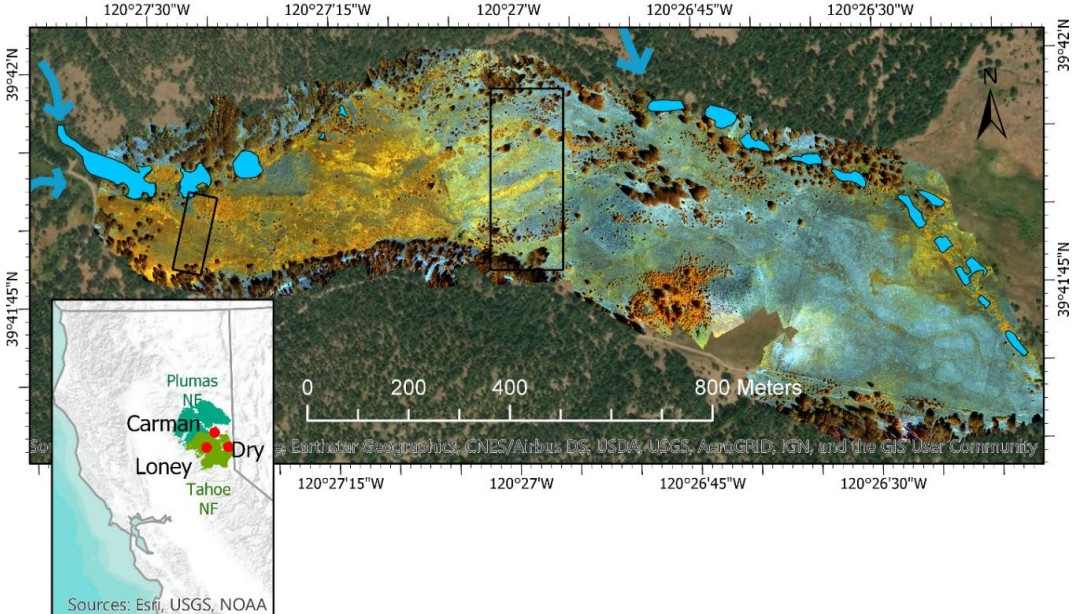

**Figure 1.** Meadow study sites. Knuthson Meadow, the largest studied on the Carman system, with July and August 2017 UAS-collected multispectral imagery from this project. False color image displays near infrared (NIR) (820–860 nm) as red, red-edge (712–722 nm) as green, and red (663–673 nm) as blue. Ponds from pond-and-plug restoration follow the pre-restoration incised channel along the northern edge of the meadow. Major surface streamflow inputs are indicated with arrows. Flow is toward the east to southeast, with the westernmost pond overflowing into small channels to the south. Locations of cross-section Figure 8 (left) and Figures 9 and 10 (right) are shown as rectangles. The inset map locates the three Tahoe National Forest meadow systems that were the focus of this study: Carman (including upper Carman Valley and Knuthson), Dry, and Loney, as well as the adjacent Plumas National Forest where numerous pond-and-plug projects have been built by the Plumas Corporation (accessed 2018).

In meadows with <40 km$^2$ watershed, scour pools typically occur upstream and downstream of intact root mats that can approximate the geomorphic function of riffles in creating roughness and grade control. Observations of flow during the spring snowmelt period demonstrates a system not unlike a pool-riffle system, with wider shallow sheet flows over the high-friction vegetated sections and concentrated flow through scoured pools (Figures 2 and 3). Meadows with gully development frequently exhibit degraded root mats which we can assume to be from a pre-incision history of wet-meadow *Carex* cover (Figure 4). Discontinuous gullies are frequently found adjacent to *Salix* copses in both Carman and Loney meadows, and commonly are associated with locally greater sinuosity, a commonly applied measure applied to stream channels [51].

The purpose of our field investigations was to capture channel development, vegetation/land cover, fine-scale elevation surfaces, seasonal hydrologic condition, and soil properties. Therefore, in-situ field sampling campaigns were augmented or combined with UAS flights employing multiple types of cameras.

During the first phase of this project, Slocombe and Davis [48] employed channel survey methods in five meadows. Plans and profiles of discontinuous ephemeral channels were surveyed using a combination of a 0.1 m accuracy GPS (for horizontal position) and <0.01 m accuracy laser level (for vertical position) in order to optimize field efficiency, while maintaining a finer level of vertical accuracy suitable for capturing accurate gradients in the 1–2% range (the laser level was also used along with a survey tape and GPS end points for a meadow cross-section in September 2018). Planform characteristics differed from alluvial systems and could reflect a unique herbaceous wetland channel habit, with biotic control having a major influence relative to fluvial effects [45–47]. Longitudinal pool morphologies exhibited headward plunge pools similar to those described by Hagberg [52] as

a dominant erosion process in the Sierra meadow streams. Bedform spacing was similar to alluvial systems, exhibiting an average pool-to-pool spacing of 6.72 channel widths, within published ranges of 5–7 for alluvial streams [53].

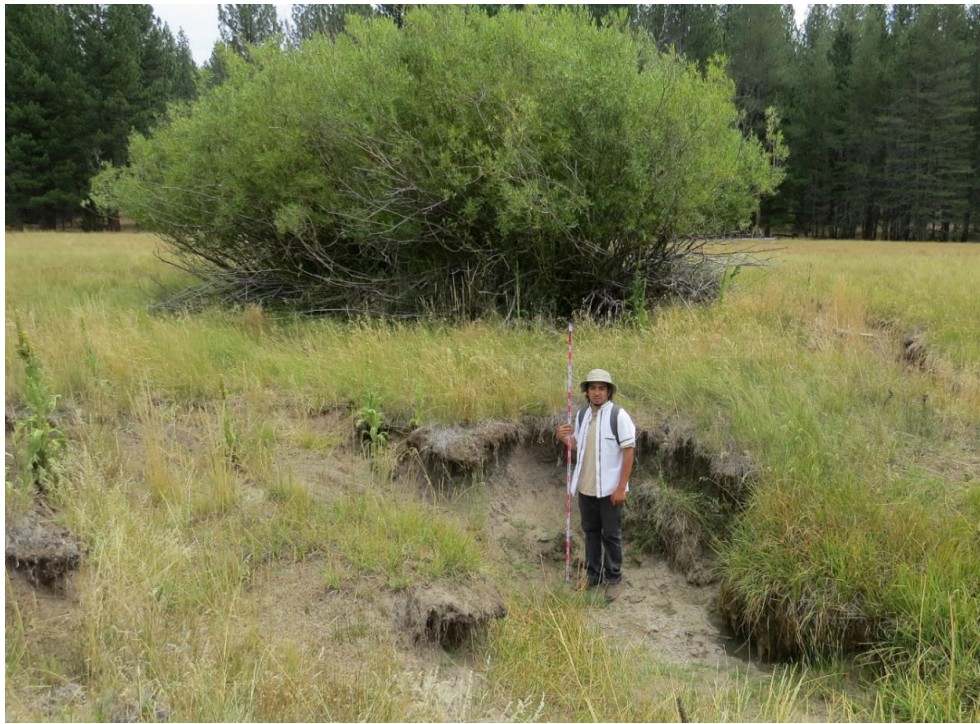

**Figure 2.** A discontinuous gully in West Fork Carman. Note *Salix lemmonii* (willow), commonly growing adjacent to scour features. Photograph is by Heather Milton, used with permission.

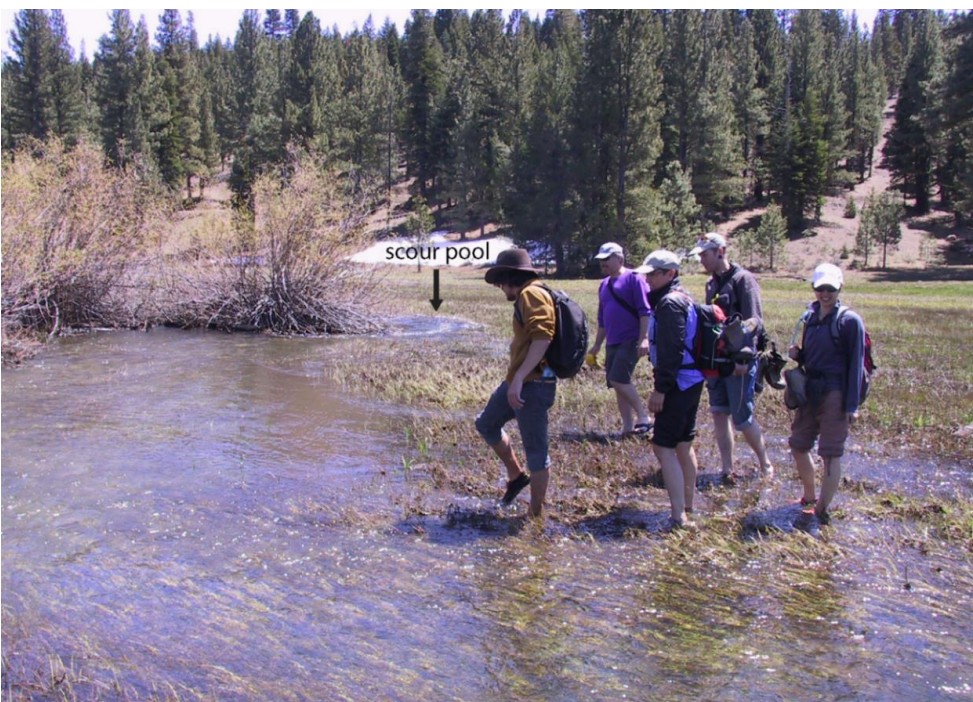

**Figure 3.** Spring snowmelt in the upper Knuthson meadow, visited during spring snowmelt on 1 May 2011, illustrating a system not unlike a pool-riffle system, with wider shallow sheet flows over the high-friction vegetated sections and concentrated flow through scoured pools, often developed adjacent to willow copses. A scoured pool is visible just to the right of willow copse at left center.

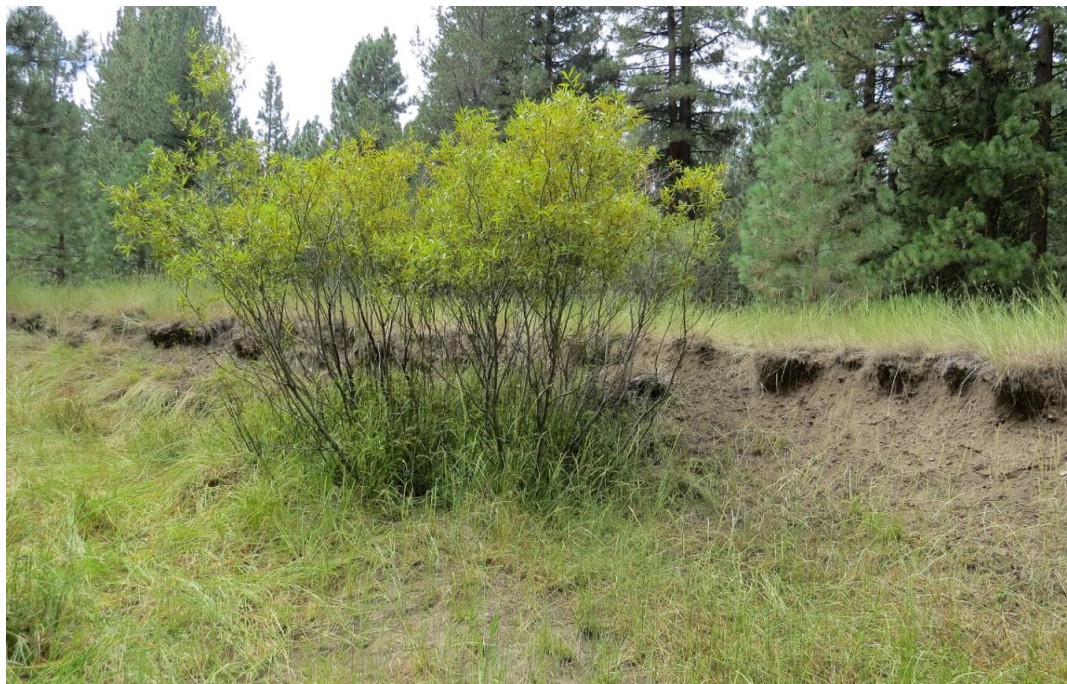

**Figure 4.** Terrace scarp exposing pre-incision (assumed) *Carex*-dominated root mat at West Fork Carman, upstream of Knuthson Meadow. Photograph is by Heather Milton, used with permission.

As part of a bio-micrometeorological investigation of degraded and restored sections of the Carman Valley system, Maher [54] sampled vegetation and soils from restored and still degraded sections, in 2014, with the restored (Knuthson) samples in the vicinity of a flux tower installation used for eddy covariance methods of assessing carbon dioxide, vapor, and energy balances [1,54]. The most hydrophilic samples were from channels ~0.5 m in depth, similar to the vegetated sections of channels surveyed in this meadow by Slocombe and Davis [48]. Vegetation quadrats (counted by Mike Vasey and Vanessa Stevens) were codominated by *Carex utriculata* and *C. nebrascensis*, whereas slightly more elevated adjacent sites had more of a mix of grasses, sedges, and rushes (with *Trifolium longipes*, *Deschampsia caespitosa*, *Juncus balticus*, *Phleum alpinum*, and *C. utriculata*, in that order, combining for 84% of quadrat species counts). The top 10 cm of soils from these sites were high in soil organic matter (SOM) and root biomass (RB), with 11.1% SOM and 4037 g m$^{-2}$ RB in the lowest elevation samples as compared with 9.9% SOM and 2438 g m$^{-2}$ RB at adjacent elevated sites. In comparison, lower SOM and RB values (4.8% SOM and 488 g m$^{-2}$ RB) were found at a degraded site further up the valley dominated by *Artemisia tridentata* and other xeric species [54].

We employed multiple approaches to capturing elevation surfaces, with the most critical being the ground terrain below vegetation. Although digital surface models (on the vegetation surface) could be captured using either color or multispectral imagery and multi-ray photogrammetry methods, the densely vegetated meadows presented challenges for capturing actual ground terrain, with small channels easily obscured. Light detection and ranging (LiDAR) data, both conventional aerial and ground-based terrestrial laser scanning (TLS), were employed, although, ultimately, the aerial methods proved to be the only way to reasonably capture sufficient bare ground, given the size of meadows and limited ability to penetrate dense herbaceous vegetation from the limited terrestrial vantage points.

Airborne LiDAR data, captured in 2013 and 2014, using an Optech Gemini Airborne Laser Terrain Mapper (ALTM) equipped with an Optech 12-bit waveform digitizer were provided by Tahoe National Forest. The average point density (under ideal conditions) was 7–8 points per square meter with a point spacing of 0.29 to 0.48 m; reported horizontal and vertical accuracies were 0.02–0.72 m and 0.05–0.35 m, respectively. Collection and processing followed the National Center for Airborne Laser Mapping (NCALM) protocols [55]. Then, we used a 1 m bare-ground digital terrain model (DTM)

interpreted from this data, set to derive elevation derivatives and topographic cross-sections (using the interpolate Shape tool in ArcGIS). Terrain analysis tools from the System for Automated Geoscientific Analysis (SAGA) were applied to a filled DTM product to derive cell-based flow accumulation (settings, top-down and multiple triangular flow direction) [56]. Then, a natural logarithmic function of the flow accumulation result was derived, which was somewhat similar to topographic wetness index (TWI), but more clearly interpreted as:

$$lnflow = \ln\left(n\left(res^2\right)\right) \qquad (3)$$

where $n$ = flow accumulation source cell count and $res$ = DTM cell resolution in m.

We experimented with multiple cameras and sensors to capture vegetation and other cover, including small cameras such as the GoPro Hero 3 and the Canon S95, mounted to a hobbyist-grade 3DR Arducopter UAS. These were used to capture standard color photography using intervalometer settings to capture photographs at various time intervals, commonly 2 s, depending on flight velocity and environmental conditions. One of our Canon S95 cameras was modified to include near-IR by removing the IR blocking filter and adding a Hoya A25 red filter to limit bands to red + IR and IR [57,58]. This system, when employing standard color, also proved to be highly successful for mapping a hillslope gully with little vegetation cover [58] and was also used for our 2013 images of Knuthson and Dry Creek (pre-restoration) meadows. The best results in meadows were achieved, in 2017 and 2018, using the MicaSense RedEdge, a 230 g multispectral sensor designed for precision agriculture, mounted on a 3DR Solo quadcopter (Figure 5), with geolocation RMSE generally below 1 m, sufficient for associating with vegetation patches and channels. This 5-camera system captured narrow bands of 465–485, 550–570, 663–673, 820–860 (NIR), and 712–722 (red-edge) nm suitable for processing in the Pix4D multi-ray photogrammetry software using its "Ag multispectral" settings, employing pre- and post-images of a calibration panel to provide reflectance in the 5 bands. The drone-mounted configuration also employed a downwelling light sensor and stored its readings with the TIFF image Exif data, which was, then, used by the software to attempt to remove the effect of varying sunlight due to cloud passage. The red-edge band combined with the NIR band has been used to detect seasonal moisture stress and species [59], as well as the chlorophyll content of crops [60] which also applies to naturally watered landscapes during the growing season.

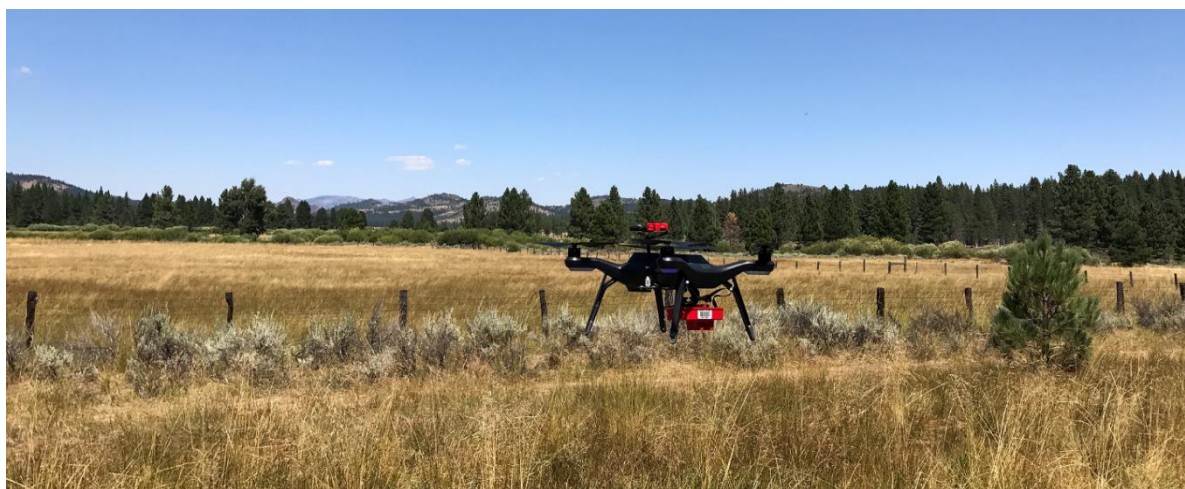

**Figure 5.** MicaSense RedEdge 5-band camera mounted on a 3DR Solo quadcopter, at lower Knuthson Meadow. This small drone (26 cm width) is capable of carrying a small camera such as a GoPro, or the 150 g MicaSense unit as shown here. Photograph is by Quentin Clark, used with permission.

We started to explore the use of long-wave IR (thermal) imagery captured by using the MicaSense Altum camera, in order to better detect patterns of soil moisture from groundwater sources, as well as channel habitat characteristics and, possibly, the assessment of hyporheic exchange [61] which is

important for freshwater ecology. For this heavier (406 g) camera, we employed a DJI Matrice 100 UAS. The camera has a similar 5-band multispectral set of sensors, i.e., 465–485, 550–570, 663–673, 712–722, and 820–860 nm, at a finer resolution, with a 2064 × 1544 sensor producing a ground sample distance of 5.2 cm at 120 m above ground level (AGL), coupled with a 160 × 120 thermal infrared sensor producing 81 cm pixels from 120 m AGL. Processing along with the 5 multispectral bands using Pix4D Ag Multispectral settings returned a geoTIFF in K.

**Table 1.** Meadows in the study. Channel widths measured from UAS imagery. Precipitation from PRISM 30-year normal model [61]. Note that the perennial Dixie Creek in Red Clover Valley is included only in our exploration of UAS-based thermal imaging.

| Meadow | Watershed Area (Km$^2$) | Mean Channel Width (M) | Elevation (M) | Annual Precipitation (Mm) | Imaging Related to Restoration |
|---|---|---|---|---|---|
| Dry | 1 | 0.90 | 1750 | 620 | before and after |
| Loney | 4 | 1.56 | 1810 | 1900 | before |
| Knuthson | 37 | 1.33 | 1520 | 800 | after |
| Carman | 15 | 1.45 | 1550 | 800 | unrestored |
| *Red Clover: Dixie Cr.* | *70* | *8.48* | *1670* | *620* | *unrestored* |

For 3DR Solo missions, flight plans were developed and deployed using the 3DR Tower app running on a Nexus 9 tablet, with along and between path settings to ensure 75% frontal and lateral image overlap; similar settings were used in Pix4Dcapture used for the DJI Matrice 100 carrying the Altum camera. In the most extensively sampled meadow, Knuthson, 14 soil and vegetation samples were collected immediately after each flight, along with 92 GPS points with linked photographs to aid in image classification, in July and August 2017; partial sections of Knuthson Meadow were also flown in September 2018 to capture senescent conditions. Basemap imagery published on Mapbox was used to allow off-line mission planning at the site. Four meadows with intermittent to ephemeral channels were mapped, including 72 ha of Knuthson Meadow, 16 ha of Carman Valley, 13 ha of Loney Meadow, and 10 ha of Dry Creek Meadow, at flight heights of 80 m (Knuthson, Dry, Carman, and Loney) or 120 m (Loney), producing a resolution of 5.6 or 8.3 cm, respectively. Maximum coverage by one flight (1 battery) was approximately 10 ha, at either 80 or 120 m altitude AGL (at ~1500 m MSL elevation). The five bands were composited to create various false-color image displays, the NDVI, and the Cl$_{re}$, derived from Equations (1) and (2). The 5.6 cm resolution composited images were also used to measure channel width, longitudinal reach length, valley length, and willow contact length (the total longitudinal channel length where the channel was in contact with willow copses), to derive sinuosity and willow influence linear proportion (*WI*), for 20 ephemeral to intermittent channels in the meadows studied, with dimensions measured using the ArcGIS Pro (2.3.3) measurement tool, where:

$$sinuosity = \frac{channel\ length}{downvalley\ length} \tag{4}$$

$$WI = \frac{\sum w}{channel\ length} \tag{5}$$

where w = segment length of willow/channel contact.

The vegetation grab samples from the three meadows were assessed in July and August 2017 to be used as ground control samples for UAS surveys, with each sample point chosen to be dominated by one species if possible, a characteristic most common with *Carex* sites. Samples from each site were identified at the species level. For 29 of these samples, the upper 10 cm of soil at point sample locations were analyzed for soil moisture and bulk density. Bulk density of the upper 10 cm of soil was assumed to be mostly influenced by root density (as roots have much lower density than inorganic soil particles), and root strength has been shown to be a major contributor to soil erosion resistance [26].

## 3. Results

The primary focus of our study is on meadows in watersheds of limited size (less than 40 km$^2$) that are drained (from a surface perspective) by ephemeral streams, in contrast to those drained by larger pool and riffle stream systems. Most meadows of this description have been subject to some form of restoration or protection method, such as exclosure fencing, check dams, and pond-and-plug methods. Other methods used effectively elsewhere, such as induced meandering [44], have not been used in most of these meadows (the perennial Dixie Creek is an exception). The meadows chosen for study (Table 1) were observed either before, after, or both before and after pond-and-plug restoration. Effective pond-and-plug restoration should show not only reduced incision but also an increased cover of hydrophilic as compared with xeric vegetation due to a raised water table.

The **Knuthson Meadow** imagery covers one of the sites used in the 2011–2012 channel study [48], and one of the sites used in the 2014 micrometeorological study with vegetation and soils sampled [54]. The channel study section is at the headmost 4 ha of Knuthson Meadow (Figure 6), a section that exhibited luxuriant growth of sedges following a wet 2017 snow year. Areas away from the channel flows are also well vegetated until the more elevated *Artemisia*-covered sites are reached, so channels are not well represented in the imagery. In contrast, the section approximately in the middle of the entire 72 ha Knuthson Meadow sampled for the 2014 micrometeorological study has clear sinuous channel zones with hydrophilic vegetation separated by more mesic to xeric zones (Figure 7). Using multiple views helps us to visualize these elusive ephemeral channels. Figure 8 combines four views of a cross-meadow elevation profile in an area with few distinct channels, at the head of a meadow where flow from upstream enters the meadow (see Figure 1), including two channels surveyed in 2012. The four views are the following: (A) false color, (B) NDVI, (C) 21 April 2013 drone color image mosaic showing surface flow from snowmelt, and (D) natural logarithm of flow accumulation over false color.

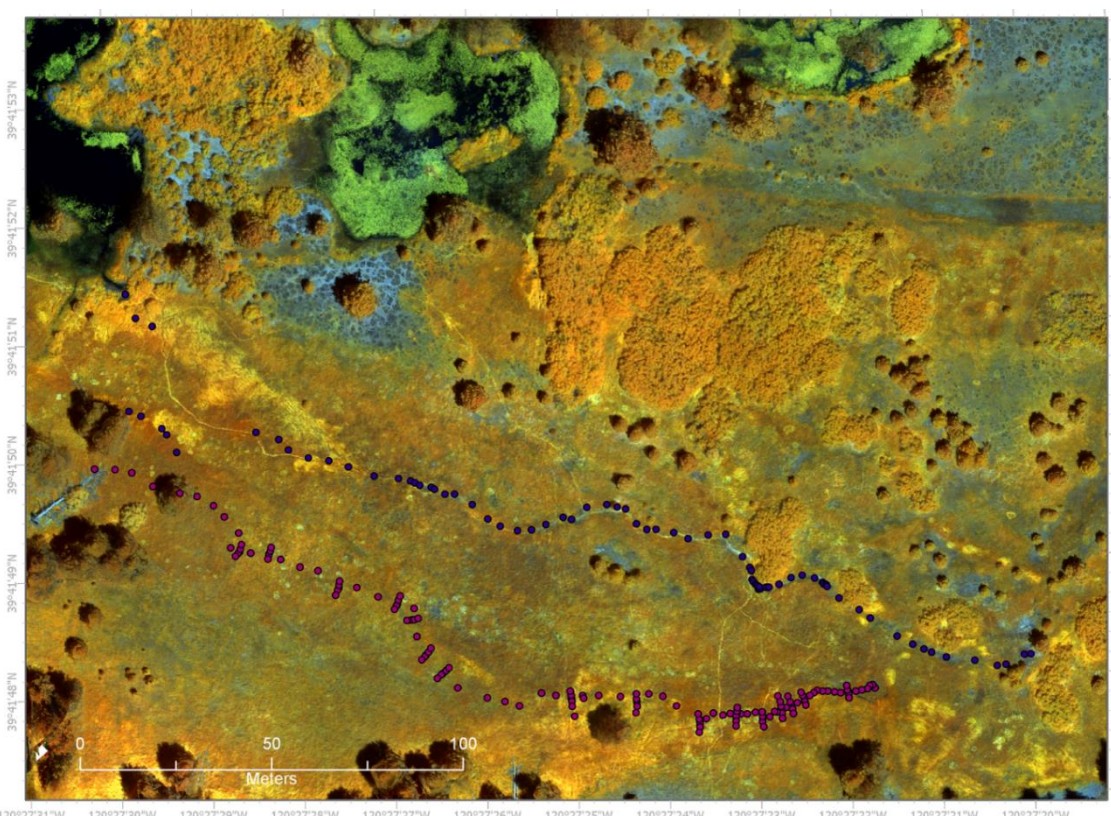

**Figure 6.** Upper Knuthson Meadow. Channel survey points surveyed by Slocombe and Davis (2014).

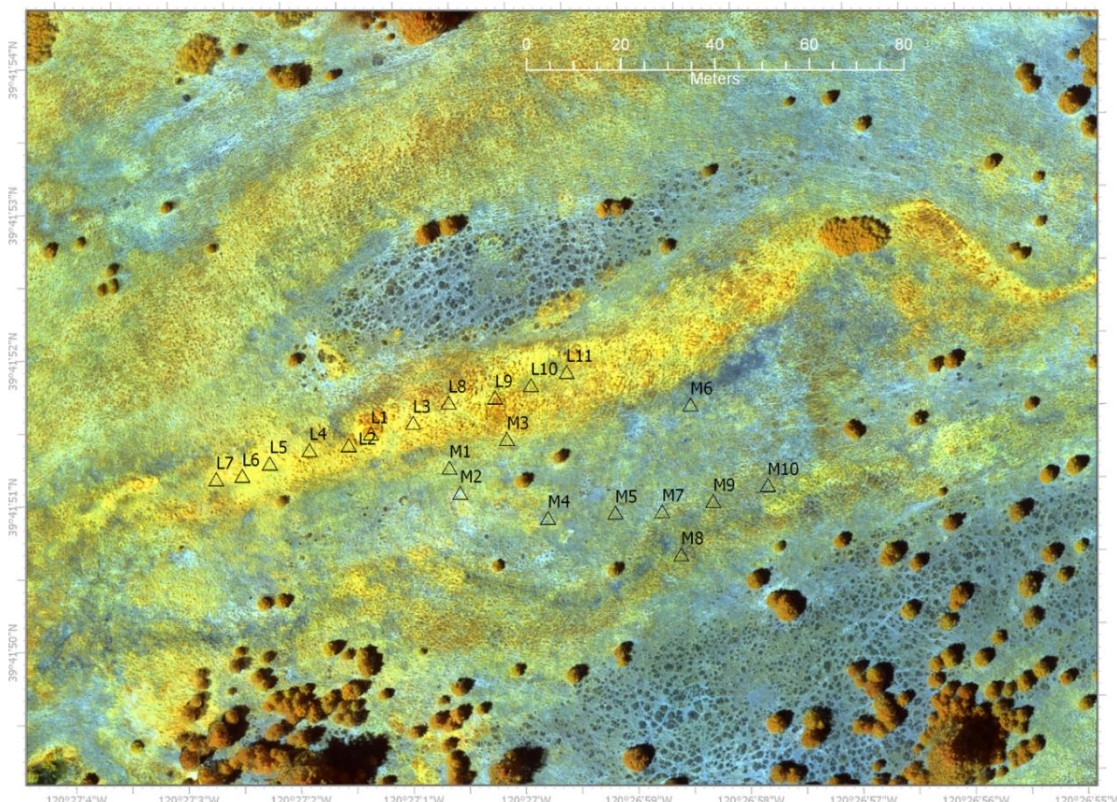

**Figure 7.** Central Knuthson Meadow. Site of an eddy covariance micrometeorological study by Maher (2016). The 21-vegetation quadrat and soil sample locations used in that study are shown as point symbols, with 11 "L" sites located in lower areas dominated by *Carex* and 10 "M" sites on slightly elevated (+ ~0.5 m) sites with more diverse facultative wetland and upland herbaceous species.

Channels were detected in Knuthson Meadow by either direct observation, survey, flow accumulation, LiDAR cross-sections, or vegetation patterns. These channels varied from features with periodic pool scouring to no scouring at all. A laser-level cross-section was surveyed at the mid-Knuthson site, allowing for a more accurate assessment of the extent and depth of channel development and its vegetation relationship. Figure 9 shows the area of the meadows sampled in 2014 in three views as follows: (A) growing season false color, (B) senescent-season false color, and (C) flow accumulation. In this area, little scouring is apparent along the channel sampled in 2014, however, the trend of concentrated flow is clearly seen by comparing the cross-section to the bright yellow and orange areas of *Carex* domination, which also follows a high flow-accumulation path. A string of *Salix* copses is visible in the imagery at about 30 m in the cross-section, to the north (left in the image) of the sampled channel, and a few scour features are evident in the imagery adjacent to copses.

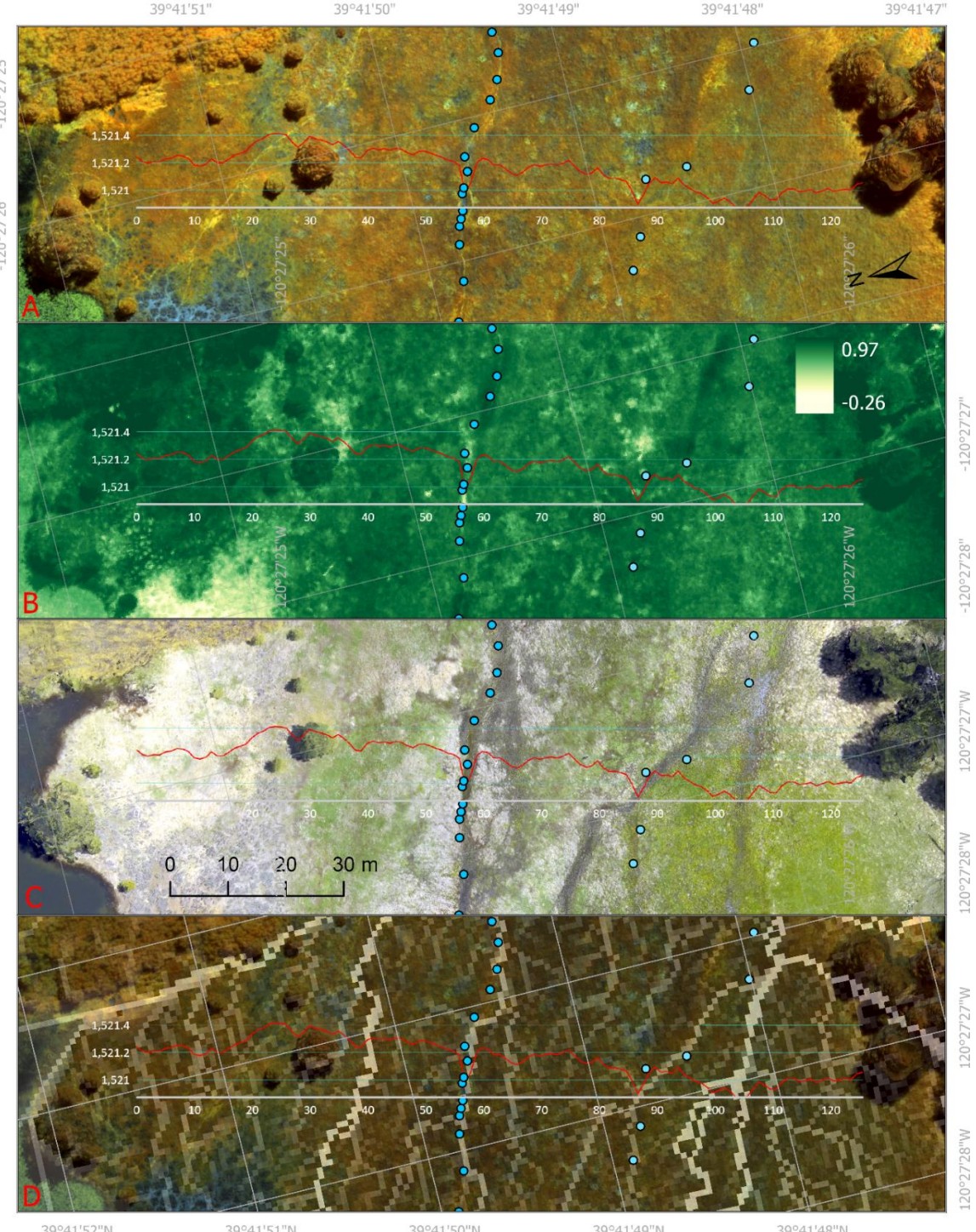

**Figure 8.** Upper Knuthson cross-section. Displayed as (**A**) false color, (R = NIR, G = red-edge, B = red); (**B**) normalized difference vegetation index (NDVI); (**C**) 21 April 2013 drone color image (GoPro Hero 3) mosaic; and (**D**) natural logarithm of flow accumulation over false color. The cross-section was derived along the white line with distances in meters from Tahoe National Forest LiDAR. The head-most pond of the pond-and-plug restoration is partially visible with the bright green color of aquatic vegetation at the letter A, while a small area of sagebrush is immediately to the south.

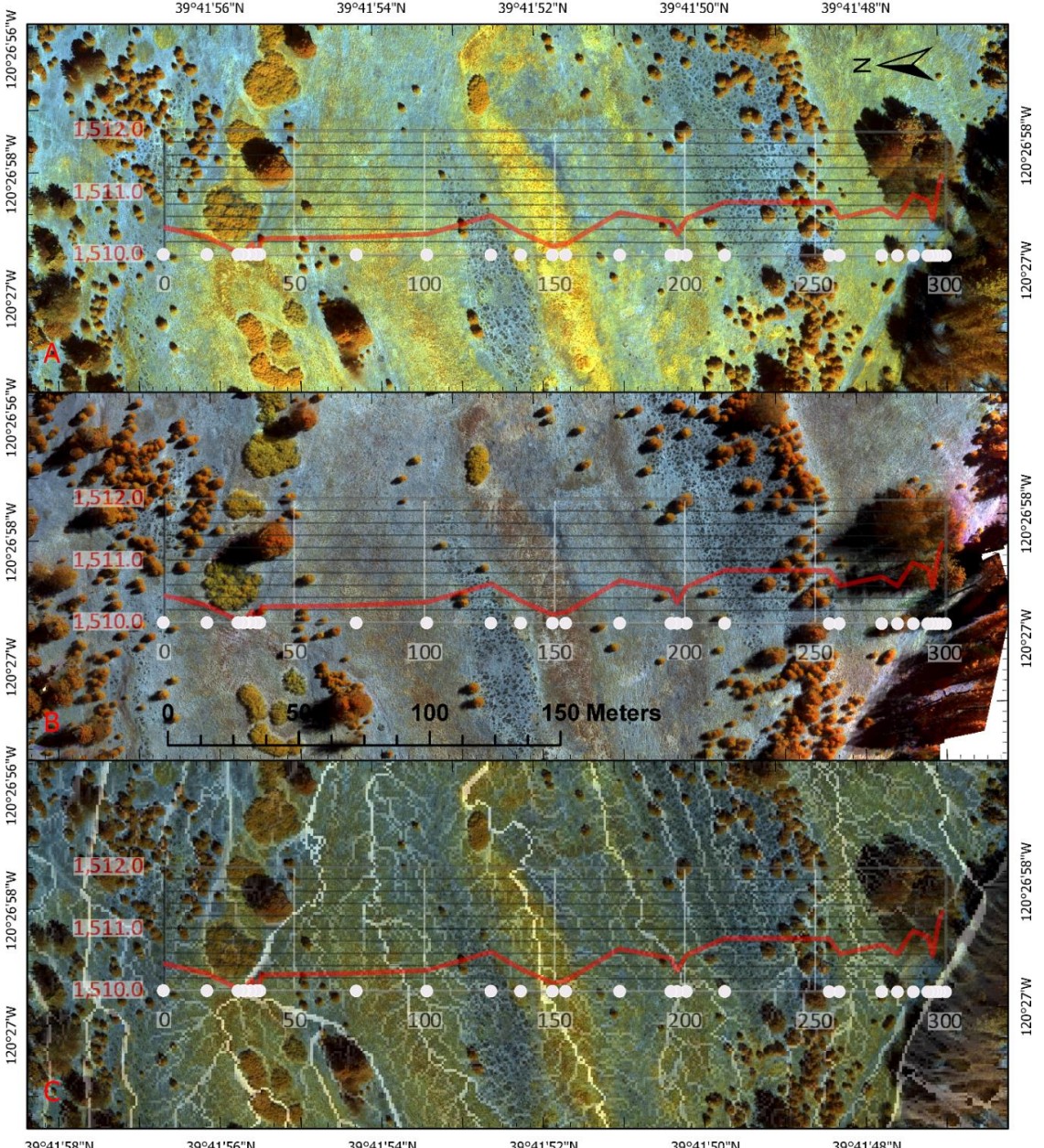

**Figure 9.** Central Knuthson Meadow cross-section. Laser-level surveyed cross-section superimposed on UAS images from growing and senescent seasons. (**A**) Growing season July 2017 false color (NIR/red-edge/red as RGB); (**B**) Senescent season September 2018; and (**C**) Natural logarithm of flow accumulation over growing season image. False color image displays NIR (820–860 nm) as red, red-edge (712–722 nm) as green, and red (663–673 nm) as blue. Cross-section units are meters horizontally and vertically.

UAS imagery collected on 23 September 2018, not surprisingly, exhibits highly senescent (and grazed) meadow vegetation (Figure 9B). The NDVI and $Cl_{re}$ indices from both phenological periods provide a view of the likely effect of topographically related soil moisture variations in influencing tendencies ranging from xeric *Artemisia* to hydrophilic *Carex*-dominated sites (Figure 10). During the senescent phenological phase, NDVI and $Cl_{re}$ are greatly reduced, with high values only in trees, although drainage lines such as the *Carex*-dominated area at cross-section distance 150 m has a higher NDVI than more elevated mixed graminoid or sagebrush-dominated areas. Leaving out the trees, there is a clear pattern seen in Figure 11 of higher NDVI and $Cl_{re}$ values in lower-lying areas

(Table 2), but, not surprisingly, both indices are significantly lower (NDVI t-test $p = 1.048 \times 10^{-11}$ and $Cl_{re}$ t-test $p = 7.541 \times 10^{-9}$) in the senescent period later in the season. Before sampling for this analysis, in order to reduce noise from vegetation texture and shading effects, the NDVI and $Cl_{re}$ rasters were derived as $5 \times 5$ focal means of $3 \times 3$ aggregate cells from the original 5 cm data.

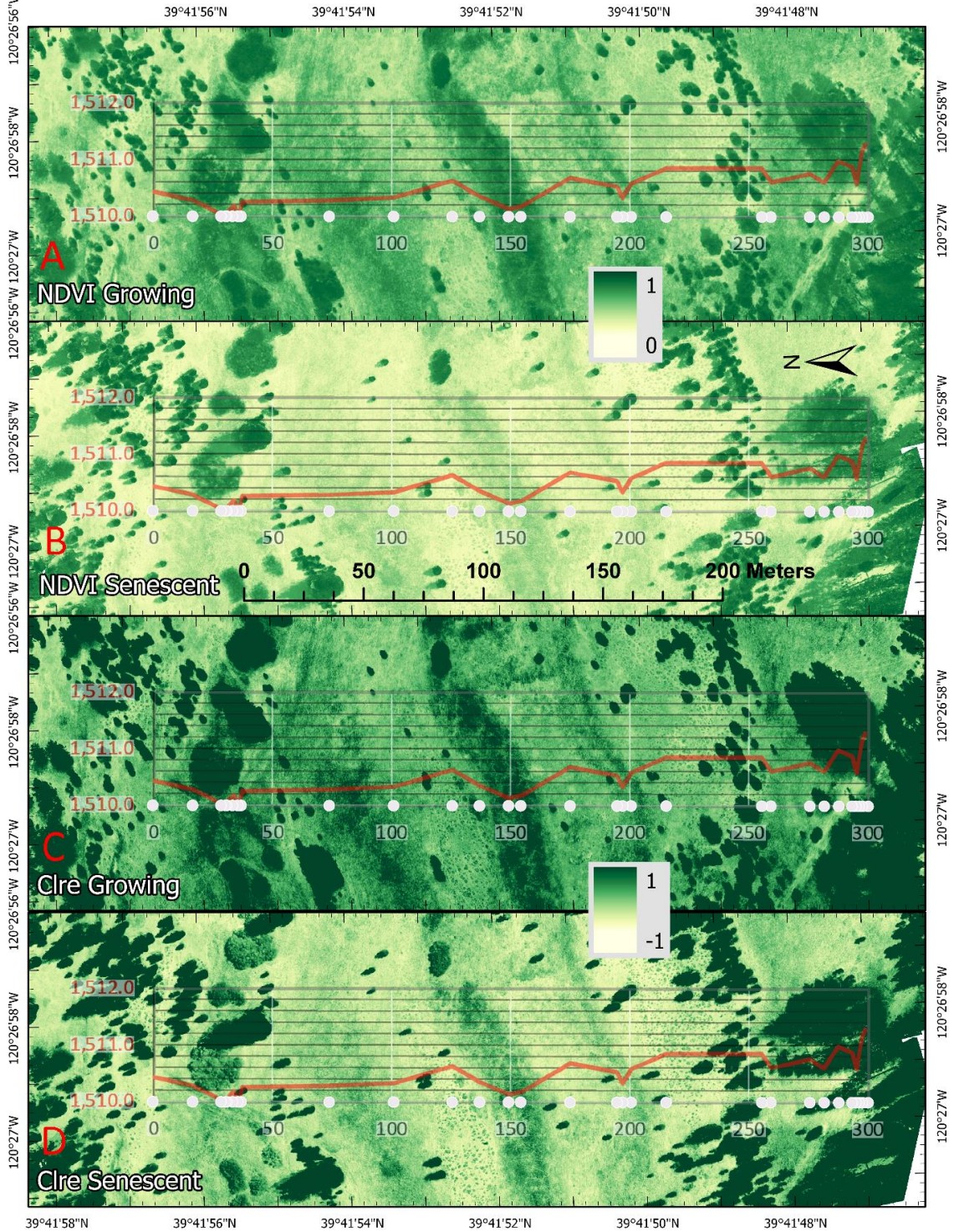

**Figure 10.** NDVI and $Cl_{re}$ indices of central Knuthson Meadow cross-section. Same cross-section as previous figure. (**A**) July 2017 NDVI; (**B**) Sep 2017 NDVI; (**C**) July 2017 $Cl_{re}$; and (**D**) Sep 2018 $Cl_{re}$. Legend corresponds to all maps.

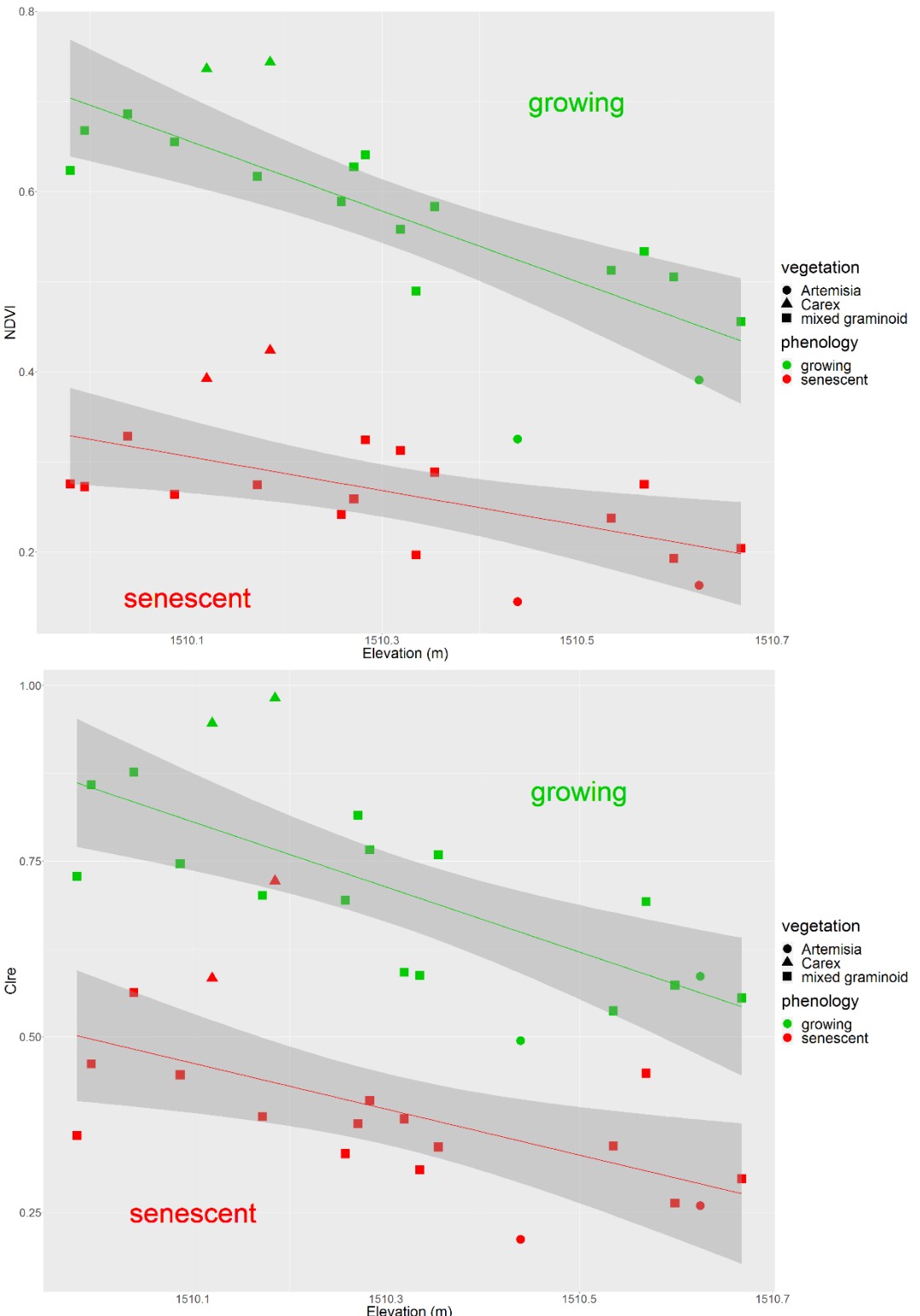

**Figure 11.** NDVI and Cl$_{re}$ during growing and senescent seasons, plotted at elevations of non-tree points along the 23 September 2018 laser-level cross-section. Low-lying areas in closer contact to groundwater sources are indicated by lower elevations.

**Table 2.** Linear model results of the NDVI and red-edge chlorophyll index ($Cl_{re}$) predicted by elevation at vegetation samples along the north-south leveling transect in Figure 10. Tree (willow and pine) samples are excluded.

| Model | Slope | $r^2$ | $p$ |
|---|---|---|---|
| lm(NDVIgrowing ~ Elev) | −0.39 | 57% | 0.00012 |
| lm(NDVIsenescent ~ Elev) | −0.20 | 31% | 0.00741 |
| lm($Cl_{re}$growing ~ Elev) | −0.46 | 48% | 0.00063 |
| lm($Cl_{re}$senescent ~ Elev) | −0.33 | 29% | 0.01019 |

As has been commonly noted in other studies [23,40], groundwater is a significant part of meadow hydrology, and pond-and-plug restoration has had a major influence on both seasonal water table fluctuations and evapotranspiration from pond surfaces and vegetation. Knuthson Meadow was classified by Rodriguez et al. [40] as exhibiting properties of a "valve"-type conceptual model, since it experiences minimal seasonal pond decline, expected to have resulted from groundwater discharge which maintains pond levels despite seepage and evapotranspiration. We can see evidence for both influx and efflux in Knuthson Meadow. By far the largest flow entering Knuthson Meadow comes from the 30 km$^2$ basin to the northwest, entering the westernmost pond, but greater surface flow is evident from spring snowmelt from the 3 km$^2$ basin to the southwest (Figure 12). In fact, recharging the meadow aquifer is a goal of pond-and-plug meadow restoration, with most ponds intended to be controlled by groundwater and not significantly involved in surface channelized flow; this northwest pond would appear to be a site of significant groundwater influx, by design. In other parts of the meadow, groundwater efflux is evidenced by redox conditions, where there are either geologic spring sources or through-meadow resurgences.

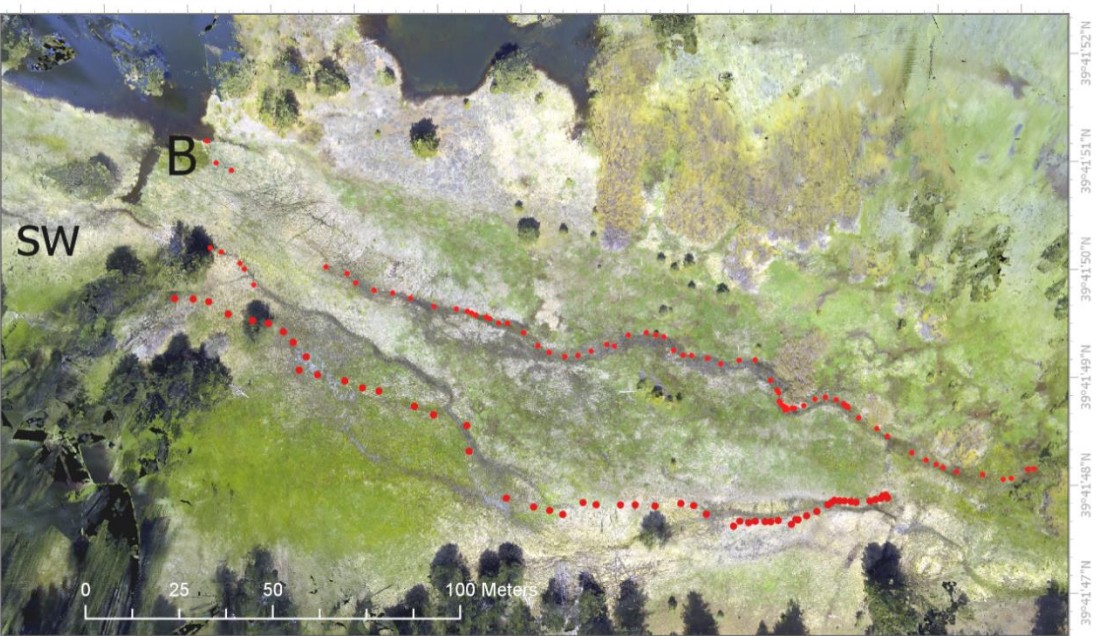

**Figure 12.** West end of Knuthson meadow from a 21 April 2013 color image mosaic. At this time, surface flows can be seen in ephemeral channels. The largest inflow into the meadow is into the pond above (**B**) in the image, with a basin area of 30 km$^2$; at (B), beavers have frequently built dams to increase the height of the pond. Flow from the southwest (SW) is from a much smaller basin of 3 km$^2$, yet this contributes more evident surface flows. Channel surveys from 2012 are shown. Imagery was captured using a 3DR Arducopter-mounted GoPro Hero-3 camera and processed with Pix4D. Distorted results in the forest surrounding the meadow is to be expected given limited image coverage and should be ignored as the focus is on the meadow.

**Loney Meadow** also illustrates a complex set of channels. In the cross-section (Figure 13) of the meadow relatively near its western (downstream) end, several areas of focused flow can be seen in the log(flow) view. The deepest channels are not necessarily the wettest predicted by flow accumulation, nor are they observed in the field or in imagery to have the greatest flows. This likely resulted, in part, from the effect of 20th century ditching along the south side of the meadow; however, groundwater flow interactions are likely, as seen in the patterns of vegetation and NDVI and $Cl_{re}$ that do not closely match the LiDAR-derived flow accumulation results. Salix copses are evident in the deepest channels and are associated with the greatest local sinuosity.

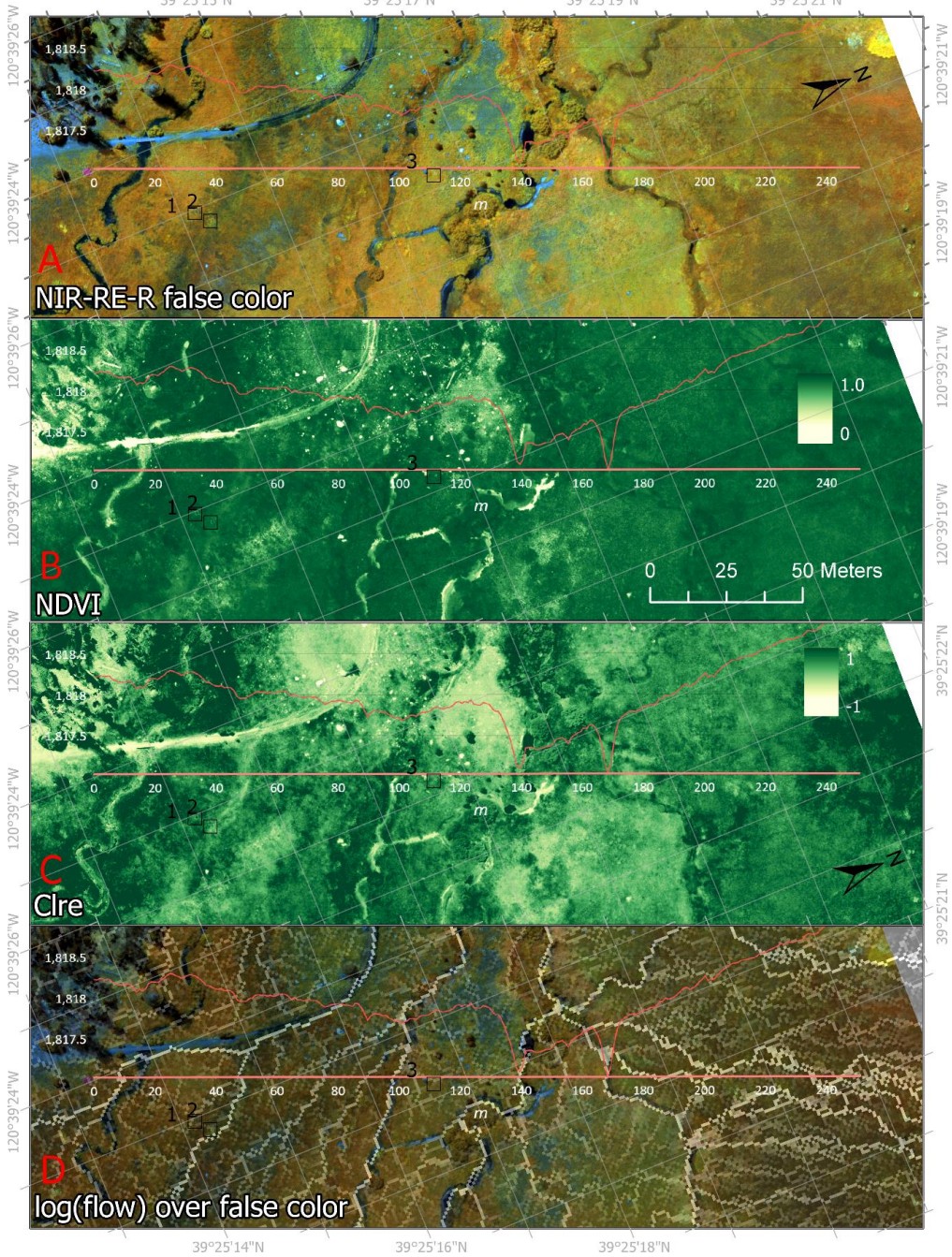

**Figure 13.** Loney Meadow cross-section. Four panels of the same areas display (**A**) NIR/red-edge (RE)/red (R) false color, (**B**) NDVI, (**C**) $Cl_{re}$, and (**D**) natural logarithm of flow accumulation over false color. The cross-section was derived along the orange line. Note the main trail and bridge crossing the meadow in the northwest corner of the image.

**Dry Creek Meadow** was flown before and after pond-and-plug restoration and exhibited changes from hydrophilic plants only along inset floodplains to more widely distributed patches (Figure 14). Although the Canon S95 camera was altered to capture NIR, it was not capable of producing a true NDVI, and an approximation was sufficient to clearly detect hydrophilic vegetation growing in a narrow inset floodplain [57], in July 2013. In the July 2017 imagery using the MicaSense RedEdge camera, much greater coverage of hydrophilic plants including *Carex* species demonstrated the effectiveness of the restoration project. This tendency was also observed in the field by Tahoe National Forest hydrologist Randy Westmoreland, who accompanied the UAS team and helped plan our field data and imagery collection. The extensive distribution, away from observed channels, supports a hydrologic system with a significant groundwater component, the goal of the restoration. Generalized to the entire meadow, an increase in NDVI can be seen, a result that was observed from Landsat imagery in 30 of 31 restored meadows studied in the Sierra Nevada by Rodriguez et al. [40].

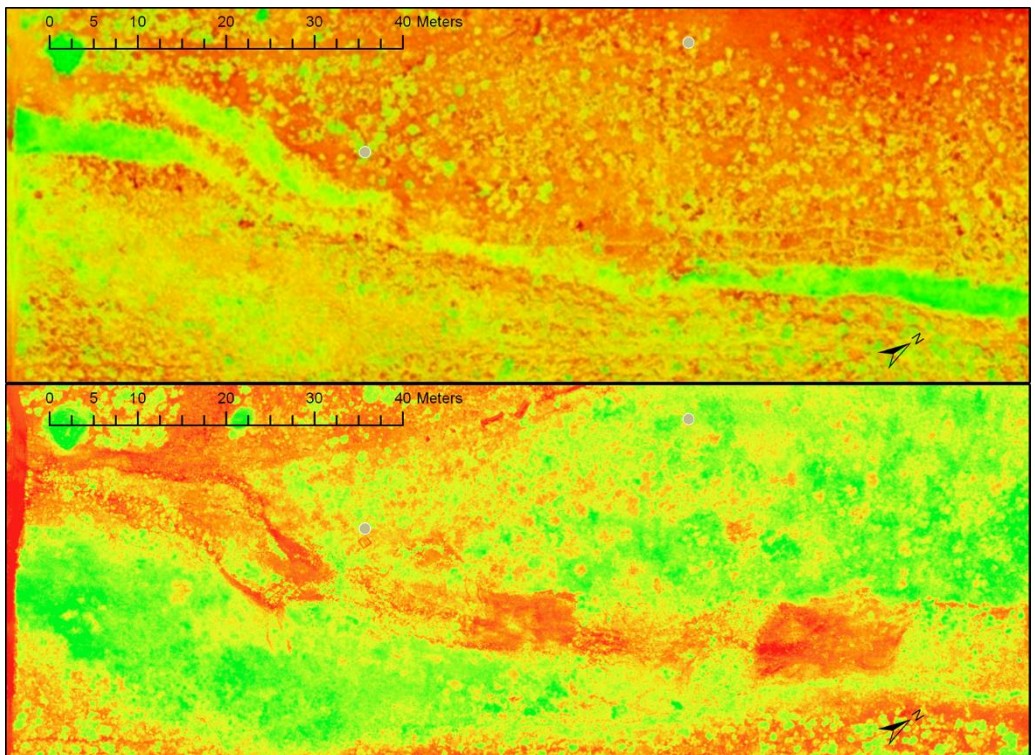

**Figure 14.** NDVI of part of Dry Creek Meadow. Collected 26 July 2013 (top) and 23 July 2017 (bottom), with bright green representing high NDVI values. In 2013, hydrophilic plants producing high NDVI values are only found along an inset channel floodplain. This channel was the focus of pond-and-plug restoration, which raised the water table sufficiently to create much more extensive hydrophilic growth indicated by high NDVI values in the 2017 map. The 2013 map is modified from Christian [55].

*Analysis: Sinuosity, Vegetation, and Soil Properties*

The relationship of sinuosity to willow influence observed above was assessed using measurements from imagery where the 5 cm resolution composite images accurately detected channel widths, especially where scoured. Detection of ephemeral channels in graminoid-covered reaches was difficult in the field and, of course, also proved difficult in imagery. However fine details of vegetation patterns could often be used, and where these were consistent with upstream or downstream scour features, they were able to guide channel width interpretation reasonably well, although we were more confident in channel length and willow contact length, and thus sinuosity and willow influence (WI) as a linear proportion (Equation (5), Figure 15). A linear model of sinuosity~WI appears to be significant, with $p = 0.00336$. The $r^2 = 0.3537$ is reasonably high considering that many other factors are

going to influence sinuosity, such as historical changes in vegetation and the effect of sheep (Carman, Knuthson) or cattle (Loney, Dry) grazing.

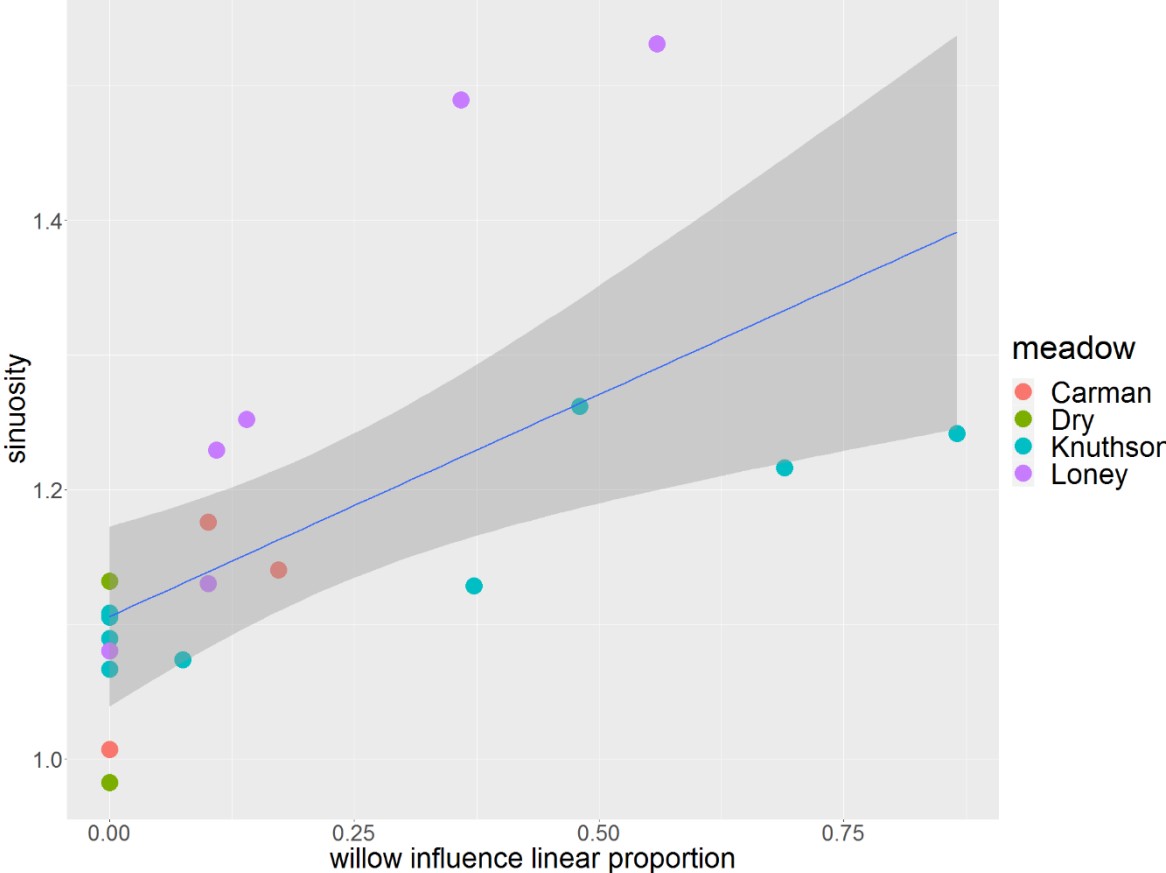

**Figure 15.** Sinuosity in relation to willow influence (WI), the linear proportion of channels that are in contact with willows, $p = 0.00336$ and $r^2 = 0.3537$.

From soil samples of meadows studied (Table 3), not including Red Clover, bulk density and soil moisture are the key results, as is NDVI detected in imagery at those point locations. Low bulk density of the upper 10 cm of soil is especially notable in the dense root mats of sedges and rushes. Plant species were grouped into one of four categories, three of which were wetland associated and one upland. The *Carex* group of species included *Carex nebrascensis*, *C. utriculata*, and *C. pellita*, all of which are wetland obligate species characterized by dense root mats. The *Juncus* species are *J. balticus* and *J. nevadensis*. Sites with grasses involved genera such as *Deschampsia* and *Danthonia,* as well as meadow forbs such as *Symphiotrichum*. Upland species comprised a range of shrubs including the xeric *Artemisia tridentata*. While all meadows also included scattered willow copses and pines, these were not sampled, although GPS-tagged photographs were used to link these trees with patches in the imagery. For bulk density, while the four groups are not significantly distinct in an ANOVA test, *Carex* species vs. all other samples are distinct ($p = 0.0342$). Soil moisture in July samples is significantly distinct for all groups ($p = 0.0143$).

Additional vegetation-only samples were collected as GPS points with captured photographs; 18 observations with vegetation types easily identified in the photographs were used and added to the above vegetation and soil sample locations. For a total of 38 vegetation samples and observations, NDVI values were extracted from the UAS-collected multispectral image data from July 2017 (see Table 2). For NDVI, the four vegetation groups are significantly distinct in an ANOVA test ($p = 0.009$) and *Carex* is also distinct from all other vegetation types ($p = 0.045$).

**Table 3.** Soil and vegetation sample results. Significant grouping variables by ANOVA are given with asterisks. Vegetation groups are significantly distinct based on mean soil moisture ($p = 0.0143$) and NDVI ($p = 0.009$), while only *Carex* is distinct from all other groups based on bulk density ($p = 0.0342$). The *Carex* group is represented only by 3 hydrophilic, wetland obligate species *C. nebrascensis*, *C. utriculata*, and *C. pellita*. *Juncus* species are also wetland *J. balticus* and *J. nevadensis*.

| Vegetation Group | Bulk Density | | Soil Moisture % (July) * | | NDVI (July) * | |
|---|---|---|---|---|---|---|
| | mean | s | mean * | s | mean * | s |
| *Carex*-dominated | 0.629 * | 0.258 | 49.7 | 19.0 | 0.860 | 0.052 |
| *Juncus*-dominated | 1.308 | 0.996 | 26.2 | 17.0 | 0.669 | 0.146 |
| grasses/mixed | 1.250 | 0.773 | 19.5 | 11.9 | 0.706 | 0.167 |
| upland | 1.422 | 0.828 | 11.9 | 9.3 | 0.531 | 0.180 |

## 4. Discussion

In this study of meadows in Tahoe National Forest, the association of wetland-obligate sedges (*Carex* species) with low-lying areas of the meadow agrees with what has been observed in previous studies [22,62], and an increase of these hydrophilic plants is a likely contributor to increased NDVI in restored meadows. The methods employed in this research confirmed the observation of these effects at the fine scale of ephemeral channels that we monitored. The results from the soil samples demonstrated that for sedges, bulk density and soil moisture are clearly distinct from other vegetation types. Sedge domination in some areas clearly defined channelized flow, although this was most apparent in sections of meadows more distant from groundwater source influxes, especially noted in the central Knuthson Meadow section. Here, greater variability of micro-elevation and possibly piezometric surfaces could occur, creating a more distinct vegetation signal (see Figures 7 and 9). Evidence for a history of wetter meadow conditions could also be seen in relict root mats, although the nature of this phenomenon requires further study.

Aerial LiDAR at a sufficient density such as 7–8 points/m$^2$ of the Tahoe National Forest acquisition appeared to be suitable for detecting surface flow patterns across these meadows, although the contribution of groundwater could not be detected from elevation, and surface–aquifer interactions were clearly a major component of meadow hydrology, and thus channel development.

Capture of vegetation patterns using UAS methods designed for precision agriculture is promising and should aid in assessing the signal of restoration success. Hydrophilic vegetation depends on a high-water table, and xeric vegetation dies back where it gets too wet; these are readily identified in high-resolution, multispectral imagery captured by UASs. Where differences in vegetative condition are more clearly evident (see Figures 7 and 9), channel patterns are similar to what had been noted in channel survey studies, with low sinuosity as compared with larger alluvial channels. Although not a part of the primary research reported here, we also explored longwave IR imagery to help discern moisture patterns from their thermal signal, and these could also aid in channel studies, as apparent in an image of Dixie Creek in Red Clover Valley (Figure 16). This section of creek is currently part of the Clover Valley Ranch restoration project of The Sierra Fund [33,63], involving restoration methods ranging from grade control structures to beaver dam analogs. In contrast to the meadows with ephemeral streams that are the focus of this study (see Table 1), this meadow is drained by larger perennial streams. The thermal (LWIR) images at 81 cm resolution identify areas of the meadow with cooler temperatures which are possibly related to groundwater sources, but also clearly delineate the stream course even when otherwise obscured by riparian vegetation. Other LWIR sensors appearing on the market promise to extend these benefits to finer scales.

Willows (*Salix* species) were noted adjacent to many discontinuous gullies in both the Carman system (upper Carman Valley and Knuthson Meadow) and Loney Meadow, and willows were also associated with relatively sharp bends in the channel, producing high local sinuosity (see Figures 6, 7 and 13), as was supported by a linear model (see Figure 15). In unrestored meadows, i.e., Carman Valley and Loney Meadow, willows can grow along channels in inset floodplains (see Figures 2, 4 and 13).

Somewhat unclear at this stage is whether these shrubs preferably propagate along these sites or if their root systems or shading enhance scouring as has been documented elsewhere [27,64], whereas Stokes and Cunningham [65] noted that, in Australia, (invasive) willows spread more rapidly in streams of greater sinuosity where bare soil exposures were more common, at least at their sites. Certainly, a more incised stream with greater stream power can also more readily distribute willow stems for vegetative reproduction; willows are well known for vegetative propagation along stream courses, a property that makes them a favorite for stream riparian restoration. Sedge growth can also be affected by shading and moisture uptake by willow shrubs that create an environment conducive to root mat disturbance. Positive feedback could be contributing, with willows (once established and possibly increasing scour) creating conditions for further propagation; we could be seeing this in a string of willows at 39°41′55″ N and 120°27′00″ W (see Figure 1), which also exhibits evidence of scour.

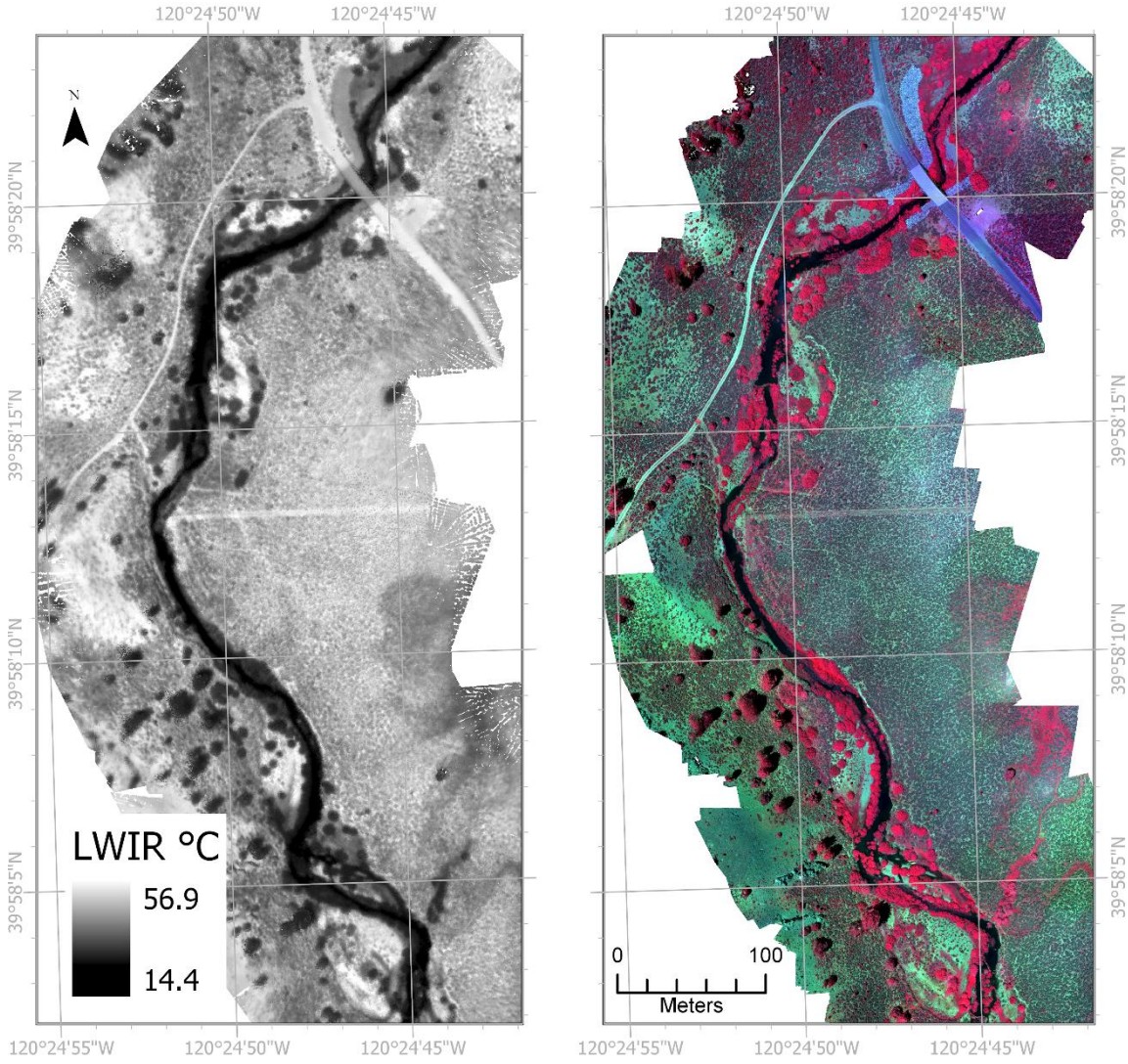

**Figure 16.** Longwave IR (**left**) and NIR-R-G false color (**right**) images of Dixie Creek in Red Clover Valley, flown using the MicaSense Altum camera mounted on a DJI Matrice 100 drone on 30 June 2019.

## 5. Conclusions

Although UAS color imagery can detect ephemeral channels, the use of multispectral cameras adds the important groundwater signal provided by hydrophilic plants. There is a clear association of hydrophilic vegetation with low-lying areas of meadows that are the sites of greater surface flow, and this vegetation is detectable in UAS multispectral imagery. For a healthy wet meadow, surface

flow is likely a minor contributor, and therefore flow accumulation based on LiDAR elevation data is insufficient to explain vegetation patterns. Groundwater is key to vegetation in these meadows, which is why a major goal of restoration is to raise the water table. This is in contrast to degraded meadows where incised channels take a major amount of the flow, and flow in these channels can be easily predicted using elevation-based flow accumulation models. If meadow restoration, then, leads to sedge growth at the expense of xeric sagebrush species such as *Artemisia tridentata* and mesic grasses typifying degraded meadows, root mats expand and can control incision. The contribution of willow is less clear, although there is an association of willow copses with scours and discontinuous gullies, often associated with an increase in local sinuosity. Capture of vegetation patterns using UAS-mounted multispectral sensors, especially coupled with aerial LiDAR, is promising for detecting vegetation and scour patterns, and should aid in assessing restoration success, and help us understand these important biogeomorphic systems.

Finally, UAS-based multispectral imagery appears to be a capable contributor to the range of remote sensing and field methods that help us to monitor the success of wet meadow restoration. Direct field measurements from topographic survey, process observation, and sampling provide the most precise and localized measurements of the hydrological, geomorphological, micrometeorological, edaphic, and physiological conditions, as well as ground control for imagery collection. Free satellite imagery, such as Landsat and Sentinel-2, provides easily accessed and frequent multispectral data, and therefore provide a meadow-scale picture of vegetation and its role as a groundwater signal, with resolutions as fine as 10 m. Recent sensors such as Planet Lab's CubeSat configurations extend four-band imagery to the order of 3 m. UAS options bridge these two ends of the detection spectrum, with five-band 8.5 cm multispectral imagery of a 10 ha meadow captured in an eight-minute flight (and with an Altum on a DJI Matrice 100, we mapped 18 ha at 5 cm resolution in 20 minutes). Thermal information from LWIR is now available and accessible from UAS platforms, and sub-meter resolution from 120 m AGL can contribute to assessing restoration success, as well as channel habitat characteristics. Field effort should, however, be considered, as each flight takes careful planning, good weather conditions, and the establishment of ground control. Depending on UASs for large study areas (we are currently working in a 1000 ha meadow) may not always be advisable, especially if fine-scale channel measurements are not needed across the study area. Combining field measurements, UASs, and satellite imagery are likely the best multiscale approach.

**Author Contributions:** Conceptualization, J.D., M.S., S.M., and P.C.; methodology, J.D., L.B., M.S., S.M., M.V., P.C., and P.L.; software, J.D. and P.C.; validation and formal analysis, J.D., M.S., S.M., L.B., and P.C.; investigation, J.D., L.B., M.S., S.M., M.V., P.C., and P.L.; writing—original draft preparation, J.D., M.S., S.M., and P.C.; writing—review and editing, J.D., L.B., M.S., S.M., M.V., P.C., and P.L.; funding acquisition, J.D. All authors have read and agreed to the published version of the manuscript.

**Funding:** This research was funded by internal San Francisco State University sources, including the Department of Geography & Environment, the College of Science & Engineering, the Institute for Geographic Information Science, and the Center for Computing in Life Sciences.

**Acknowledgments:** The cooperation with hydrologist Randy Westmoreland at Tahoe National Forest and Terry Benoit of the Plumas Corporation has been invaluable in understanding restoration work, goals, and the hydrologic and vegetation response; touring the Carman system with these two individuals was the spark that initiated this study. For help configuring a capable UAS design, we greatly appreciate the suggestions from Gregory Crutsinger, formerly of Pix4D. We also appreciate the review and approval of our proposal by the San Francisco State University (SFSU) UAS Review Board, accommodations provided by the SFSU Sierra Nevada Field Campus, and assistance on meadow plant species counts, in 2014, by Vanessa Stevens, working with coauthor Vasey. This research was also assisted by the following: students in Geog 643 Biogeomorphology of Sierra Nevada Streams and Meadows, Philip Lynch, Acacia Ross-Goedinghaus, Catherine McKnight, Kevin Physioc, as well as teaching assistant Robert Shortt for the July 2017 flights; Robert, Leonhard Blesius, and Quentin Clark for the August 2017 flights; as well as students in Geog 602 Field Methods in Physical Geography, led by Andrew Oliphant, and students in Geog 642 Watershed Assessment & Restoration for the September 2018 flights and survey. A Geog 642 team comprised of Raymond LeBeau, Axel Moser, Jeremiah Smith, and WaiTo Tsui, converted the laser-level cross-section to point features. We appreciate the data storage support and review of the figures by Anna Studwell, Associate Director of the Institute for Geographic Information Science. And finally, we greatly appreciate the useful suggestions provided by anonymous peer reviewers.

**Conflicts of Interest:** The authors declare no conflict of interest. The funders had no role in the design of the study; in the collection, analyses, or interpretation of data; in the writing of the manuscript, or in the decision to publish the results.

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
