# Peer review of "Unpiloted Aerial System (UAS)-Supported Biogeomorphic Analysis of Restored Sierra Nevada Montane Meadows"

_remotesensing, doi:10.3390/rs12111828_

Round 1

Reviewer 1 Report

The manuscript is well-written. The results are not revolutionary, but the article is inspiring in its biogeomorphic analysis approach to research montane meadows.

A combination of remote sensing and ground mapping methods was analyzed in the study area to increase the water level, which should lead to meadow restoration. The methods used are generally known, but I see the benefit of the manuscript in their interconnection in order to scientifically evaluate the potential of a study area.

The manuscript can be improved by a more detailed description of the methodology of the work, but even in this form the information is sufficient.
From my point of view, the manuscript is processed at a sufficient level and can be published as is.

Author Response

Point 1: The manuscript is well-written.

Response 1: Thank you. 

Point 2: The results are not revolutionary, but the article is inspiring in its biogeomorphic analysis approach to research montane meadows. A combination of remote sensing and ground mapping methods was analyzed in the study area to increase the water level, which should lead to meadow restoration. The methods used are generally known, but I see the benefit of the manuscript in their interconnection in order to scientifically evaluate the potential of a study area.

Response 2: As you observe, the interconnection is key to our study. Much of the technology was developed for precision agriculture, and there have been some developments in the ecological literature, but our methods integrate these developments and add new ones in order to advance the needs of restoration and the advancement of biogeomorphological theories, especially as connected to groundwater hydrology.

Point 3: The manuscript can be improved by a more detailed description of the methodology of the work, but even in this form the information is sufficient.

Response 3:

  • We have added to our rationale for using the red-edge chlorophyll index (Clre) based on precision agriculture literature.
  • We have defined sinuosity index and other equations used, and expanded the method description. While well-known in the geomorphology literature, sinuosity needs to be defined for this audience.

Point 4: From my point of view, the manuscript is processed at a sufficient level and can be published as is.

Response 4:  Thank you, and hopefully our revisions will reinforce this.

Reviewer 2 Report

This paper provides a valuable information on meadow restoration and/or watershed ecosystem and is expected to contribute to the progress of the field.

However, I think it is not suitable for publication in this journal because it does not provide new findings or new methods as remote sensing technologies.

Specific comments:

  1. Why did you use red-edge vegetation index in addition to NDVI?
  2. The following part (Line 482- 492) was duplicated:

The relationship of sinuosity to willow influence observed above was assessed using measurements from imagery where the 5cm resolution composite images allow accurate detection of channel widths, especially where scoured. Detection of ephemeral channels in graminoid-covered reaches was difficult in the field and of course also proved difficult in imagery, however fine details of vegetation patterns could often be used, and where these were consistent with upstream or downstream scour features were able to guide channel width interpretation reasonably well, though we are more confident in channel length and willow contact length, and thus sinuosity and willow influence as a linear proportion (Figure 15). A linear model of sinuosity~willow influence appears significant, with p = 0.00336. The r2 = 0.3537 is reasonably high considering that many other factors are going to influence sinuosity, such as historical changes in vegetation and the effect of sheep (Carman, Knuthson) or cattle (Loney, Dry) grazing.

  1. Please describe how “sinuosity” was estimated (calculated) showing the definition.

Author Response

This paper provides a valuable information on meadow restoration and/or watershed ecosystem and is expected to contribute to the progress of the field.

However, I think it is not suitable for publication in this journal because it does not provide new findings or new methods as remote sensing technologies.

Response:  We spent considerable time evaluating the best journal for this highly integrative study, and while it contributes to the fields of geomorphology, ecology and environmental management, it is at its core a methods paper and the key methods are in remote sensing. The novelty of the methods is also integrative, as well as in the unique perspective provided by UAS technology at scales appropriate for studying channels.

Specific comments:

  1. Why did you use red-edge vegetation index in addition to NDVI?

Response 1: This is now explained better in the manuscript, citing the work of Gitelson and others from 2003 and later.  The chlorophyll red-edge has worked in agricultural studies to identify chlorophyll variations and CO2 flux better than NDVI, and in our study it appears to provide advantages.

  1. The following part (Line 482- 492) was duplicated:

The relationship of sinuosity to willow influence observed above was assessed using measurements from imagery where the 5cm resolution composite images allow accurate detection of channel widths, especially where scoured. Detection of ephemeral channels in graminoid-covered reaches was difficult in the field and of course also proved difficult in imagery, however fine details of vegetation patterns could often be used, and where these were consistent with upstream or downstream scour features were able to guide channel width interpretation reasonably well, though we are more confident in channel length and willow contact length, and thus sinuosity and willow influence as a linear proportion (Figure 15). A linear model of sinuosity~willow influence appears significant, with p = 0.00336. The r2 = 0.3537 is reasonably high considering that many other factors are going to influence sinuosity, such as historical changes in vegetation and the effect of sheep (Carman, Knuthson) or cattle (Loney, Dry) grazing.

Response 2: Thank you – duplicate removed

  1. Please describe how “sinuosity” was estimated (calculated) showing the definition.

Response 3: 

This paper provides a valuable information on meadow restoration and/or watershed ecosystem and is expected to contribute to the progress of the field.

However, I think it is not suitable for publication in this journal because it does not provide new findings or new methods as remote sensing technologies.

Response:  We spent considerable time evaluating the best journal for this highly integrative study, and while it contributes to the fields of geomorphology, ecology and environmental management, it is at its core a methods paper and the key methods are in remote sensing. The novelty of the methods is also integrative, as well as in the unique perspective provided by UAS technology at scales appropriate for studying channels.

Specific comments:

  1. Why did you use red-edge vegetation index in addition to NDVI?

Response 1: This is now explained better in the manuscript, citing the work of Gitelson and others from 2003 and later.  The chlorophyll red-edge has worked in agricultural studies to identify chlorophyll variations and CO2 flux better than NDVI, and in our study it appears to provide advantages.

  1. The following part (Line 482- 492) was duplicated:

The relationship of sinuosity to willow influence observed above was assessed using measurements from imagery where the 5cm resolution composite images allow accurate detection of channel widths, especially where scoured. Detection of ephemeral channels in graminoid-covered reaches was difficult in the field and of course also proved difficult in imagery, however fine details of vegetation patterns could often be used, and where these were consistent with upstream or downstream scour features were able to guide channel width interpretation reasonably well, though we are more confident in channel length and willow contact length, and thus sinuosity and willow influence as a linear proportion (Figure 15). A linear model of sinuosity~willow influence appears significant, with p = 0.00336. The r2 = 0.3537 is reasonably high considering that many other factors are going to influence sinuosity, such as historical changes in vegetation and the effect of sheep (Carman, Knuthson) or cattle (Loney, Dry) grazing.

Response 2: Thank you – duplicate removed

  1. Please describe how “sinuosity” was estimated (calculated) showing the definition.

Response 3: See Equation 4, now added.  Measurement methods documented in the text.

Reviewer 3 Report

Summary

This is an interdisciplinary study for demonstrating to restoration managers how to employ rapid and cost effective monitoring methods for meadow restoration. The combination of remote sensing, ecology, hydrology, and geography make this a nice contribution to demonstrating the utility of remote sensing for practical applications in restoration ecology. I recommend that it is an appropriate for publication in Remote Sensing. My comments are minor and listed below.

Specific comments

  1. Line 32 – Could add concluding sentence about how this integrated UAV approach of vegetation indices would be used in monitoring by restoration mangers.
  2. Line 43 – It would be helpful for a non-specialist to give a short description of a pond-and-plug restoration.
  3. Line 48 – 55 – This could make a really nice paragraph to setup the introduction and provides context before specifics of northern Sierra Nevada meadow restorations. Suggest splitting the paragraph here or rearranging the content of this paragraph if kept as one paragraph.
  4. Line 67 – “with a transition to more hydrophilic species, a common goal of restoration,” – missing comma.
  5. Line 100 – suggest deleting “attempt to”
  6. Line 211 – missing opening bracket for reference 48.
  7. Line 259 – Is there a reference to go with this that would provide additional specifications of the instrument etc.? Perhaps a report by the Tahoe National Forest?
  8. Line 288-292 – This sentence isn’t necessary if it refers to methods for another study.
  9. Line 325 – citation for this?
  10. Line 326 – subsection titles for the results are recommended as the section is rather lengthy and would help indicate major divisions.

Author Response

Summary: 

This is an interdisciplinary study for demonstrating to restoration managers how to employ rapid and cost effective monitoring methods for meadow restoration. The combination of remote sensing, ecology, hydrology, and geography make this a nice contribution to demonstrating the utility of remote sensing for practical applications in restoration ecology. I recommend that it is an appropriate for publication in Remote Sensing. My comments are minor and listed below.

Summary Response:

Thank you.

Specific comments:

Point 1:  Line 32 – Could add concluding sentence about how this integrated UAV approach of vegetation indices would be used in monitoring by restoration mangers.

Response 1: Done.

Point 2:  Line 43 – It would be helpful for a non-specialist to give a short description of a pond-and-plug restoration.

Response 2: A short description is added here, and since the method explained in more detail later in the article, "described below" is added here.

Point 3:  Line 48 – 55 – This could make a really nice paragraph to setup the introduction and provides context before specifics of northern Sierra Nevada meadow restorations. Suggest splitting the paragraph here or rearranging the content of this paragraph if kept as one paragraph.

Response 3: Great idea, and we've now moved it to the very start of the paper, and adjusted for flow.

Point 4:  Line 67 – “with a transition to more hydrophilic species, a common goal of restoration,” – missing comma.

Response 4: Adding the comma changes the meaning of the phrase and sentence:  "with a transition to more hydrophilic species a common goal of restoration" is the right meaning, but there admittedly is an implied "being," so perhaps it's clearer to say "with a transition to more hydrophilic species being a common goal of restoration" so we've changed it to that.

Point 5:  Line 100 – suggest deleting “attempt to”

Response 5: Agreed, done.

Point 6:  Line 211 – missing opening bracket for reference 48.

Response 6: Fixed.

Point 7:  Line 259 – Is there a reference to go with this that would provide additional specifications of the instrument etc.? Perhaps a report by the Tahoe National Forest?

Response 7: Now included.

Point 8:  Line 288-292 – This sentence isn’t necessary if it refers to methods for another study.

Response 8: Perhaps, however this thermal sensor does contribute to this study in the discussion so we were asked by another reviewer to include it here.

Point 9:  Line 325 – citation for this?

Response 9: Yes, assuming this is referring to soil erosion resistance, there's a very good reference by Micheli & Kirchner, now cited here.  For bulk density related to roots vs mineral soil, this has long been established based on the relative density of woody vs mineral soil constituents, so maybe doesn't need and probably can be assumed to be understood without citing, but we've added an explanation.

Point 10:  Line 326 – subsection titles for the results are recommended as the section is rather lengthy and would help indicate major divisions.

Response 10:  We considered various ways of doing this, and to help we moved one part to the discussion (thermal), and added a subsection heading to the general meadow analysis of sinuosity, vegetation and soils.  We hope this helps.